# East Greenland ice core dust record reveals timing of Greenland ice sheet advance and retreat

Marius Folden Simonsen [1], Giovanni Baccolo [2], Thomas Blunier [1], Alejandra Borunda[3,4], Barbara Delmonte[2], Robert Frei [5], Steven Goldstein[3,4], Aslak Grinsted [1], Helle Astrid Kjær[1], Todd Sowers[6], Anders Svensson [1], Bo Vinther[1], Diana Vladimirova[1], Gisela Winckler [3,4], Mai Winstrup [7] & Paul Vallelonga [1]*

Accurate estimates of the past extent of the Greenland ice sheet provide critical constraints for ice sheet models used to determine Greenland's response to climate forcing and contribution to global sea level. Here we use a continuous ice core dust record from the Renland ice cap on the east coast of Greenland to constrain the timing of changes to the ice sheet margin and relative sea level over the last glacial cycle. During the Holocene and the previous interglacial period (Eemian) the dust record was dominated by coarse particles consistent with rock samples from central East Greenland. From the coarse particle concentration record we infer the East Greenland ice sheet margin advanced from 113.4 ± 0.4 to 111.0 ± 0.4 ka BP during the glacial onset and retreated from 12.1 ± 0.1 to 9.0 ± 0.1 ka BP during the last deglaciation. These findings constrain the possible response of the Greenland ice sheet to climate forcings.

[1] Centre for Ice and Climate, Niels Bohr Institute, University of Copenhagen, Tagensvej 16, DK-2200 Copenhagen N, Denmark. [2] Department of Earth and Environmental Sciences, University Milano-Bicocca, Piazza della Scienza 1, I20126 Milan, Italy. [3] Lamont-Doherty Earth Observatory, Columbia University, 61 Route 9W, Palisades, NY 10964, USA. [4] Department of Earth and Environmental Sciences, Columbia University, Mail Code 5505557 Schermerhorn Extension, New York, NY 10027, USA. [5] Department of Geosciences and Natural Resource Management, University of Copenhagen, Øster Voldgade 10, DK-1350 Copenhagen, Denmark. [6] Earth and Environmental Systems Institute, 2217 EES Building, Pennsylvania State University, University Park, PA 16802, USA. [7] Danish Meteorological Institute, Lyngbyvej 100, DK-2100 Copenhagen Ø, Denmark. *email: vallelonga@nbi.ku.dk

Although ice cores are geographical point measurements, they represent a record of air, water and aerosols transported to the ice over regional or even hemispheric scales. In contrast, reconstructions of past ice sheet limits are typically limited to the locations of the individual measurements[1,2]. These measurements include dating of moraines and subglacial rocks by cosmogenic surface-exposure methods and radiocarbon dating of exposed organic material[3]. Although East Greenland is mountainous and relatively inaccessible, the deglacial timing and location of the ice sheet margin has been intensively studied, particularly in the locality of Scoresby Sund and Milne Land. It is a challenge to investigate changes in ice sheet extent prior to the LGM, due to the removal and/or reworking of chronological features such as moraines and erratics. Hence there are large dating uncertainties regarding glacial advance after the Eemian[4]. Ice core dust records may complement this research because ice caps and ice sheets are sensitive recorders of aeolian dust, such as that deflated from glacial outwash plains[5], and ice cores typically feature accurate chronologies over multimillennial timescales[6].

Records of past dust deposition have been reconstructed from central Greenland ice cores, yet no single record covers the last glacial cycle entirely. Representative dust fluxes for the Holocene have been reported as $24 \pm 9 \, \mathrm{mg \, m^{-2} \, yr^{-1}}$ from the South Greenland DYE-3 core[7,8] and Steffensen[9] reported fluxes of $7\text{–}11 \, \mathrm{mg \, m^{-2} \, yr^{-1}}$ from the central Greenland GRIP core. During the last glacial period, the ice core dust concentration was 10–100 times greater than in the Holocene due to enhanced continental aridity, increased wind strength, lower snow accumulation and longer atmospheric particle lifetime[9–11]. Around 90% of the dust mass in central Greenland during the Holocene and 95% during the glacial comes from particles smaller than 4 µm, as large particles are depleted during transport due to gravitational settling[9].

The provenance of dust in Greenland ice cores has been primarily assigned by comparing geochemical parameters with likely dust sources in arid zones of the Northern Hemisphere. Mineralogy and strontium/neodymium isotope ratios[12,13] provide the classic means of establishing dust provenance, with central Asian deserts (the Gobi and Taklamakan in particular) providing the best geochemical matches to the dust found in central Greenland during both the Holocene and last glacial. Bory et al.[14], also investigated late Holocene ice from two small coastal Greenland ice caps, Renland and Hans Tausen, identifying greater dust fluxes with distinctly less-radiogenic Sr and Nd isotope ratios (i.e. lower εNd, higher $^{87}\mathrm{Sr}/^{86}\mathrm{Sr}$) compared to central Greenland ice cores. Although no representative source was identified, Bory et al. speculated that a potential contributor was the Caledonian fold belt, a formation that comprises most of North and East Greenland[14] and for which less-radiogenic Sr isotopic signatures have been reported[15].

The dust record of the RECAP ice core was obtained from the Renland ice cap in the Scoresby Sund region of central East Greenland. The RECAP ice core (71.30°N, 26.72°W, 2315 m asl) was drilled in June 2015 less than 2 km from the site of the 1988 Renland ice core[16]. The surface elevation at the drill site is comparable to the DYE-3 (2490 m asl) and NEEM (2450 m asl) ice cores. The core reaches 584 m to bedrock, and contains a complete climate record back to 120 ka b2k (before 2000 CE) (see Supplementary Note 1, Supplementary Figs. 1–3 for information regarding the time scale). It was drilled into a topographic valley, resulting in a thick and well-resolved Holocene sequence (533 m), a strongly thinned glacial sequence (20 m) and a partially-preserved Eemian sequence (8 m) above 23 m of stratigraphically disturbed ice. The coastal location of the Renland ice cap provides important geographic climate information that can be compared with central Greenland ice cores as well as providing a

sensitive indicator of changes at the margins of the Greenland ice sheet.

We use the RECAP large dust particle record (larger than 8 µm) as an indicator of the presence of local dust sources and present new isotope geochemistry data constraining the likely sources of dust found on the Renland ice cap. The RECAP dust record also provides an independent age constraint on changes to local relative sea level and the location of the East Greenland ice sheet margin, which are both important controls on the presence of dust deflation sources through the onset of the glacial and the deglaciation. During both the Holocene and the previous interglacial period (the Eemian) the ice core dust record was dominated by coarse particles likely to originate from Kong Christian X Land in central East Greenland. We infer the East Greenland ice sheet margin advanced from $113.4 \pm 0.4$ to $111.0 \pm 0.4$ ka BP during the glacial onset and retreated from $12.1 \pm 0.1$ to $9.0 \pm 0.1$ ka BP during the last deglaciation. These findings provide important constraints for ice sheet models used to investigate the sensitivity of the Greenland ice sheet to climate forcing parameters.

## Results

**RECAP dust record.** The RECAP dust record (Fig. 1, Supplementary Fig. 4) confirms previously reported features of central Greenland dust records such as extreme concentration variability during stadial/interstadial cycles. The dust record also displays unusually high concentrations during the interglacials. Dust records from central Greenland ice cores (DYE-3, GRIP, GISP2, NGRIP) consistently feature low concentrations in the Holocene ($<10^2 \, \mathrm{µg \, kg^{-1}}$) and high glacial stadial concentrations ($>10^3 \, \mathrm{µg \, kg^{-1}}$)[17]. In contrast, the RECAP dust record displays intermediate concentrations during the Holocene (5–11.7 ky b2k, $305 \pm 117 \, \mathrm{µg \, kg^{-1}}$, 1σ error bounds) and late Eemian (119.0–120.8 ka b2k, $861 \pm 402 \, \mathrm{µg \, kg^{-1}}$, 1σ error bounds) with higher concentrations in the glacial stadials ($>10^3 \, \mathrm{µg \, kg^{-1}}$) and lower concentrations in the glacial interstadials ($<200 \, \mathrm{µg \, kg^{-1}}$). We evaluate the glacial and interglacial sections of the RECAP dust record separately below.

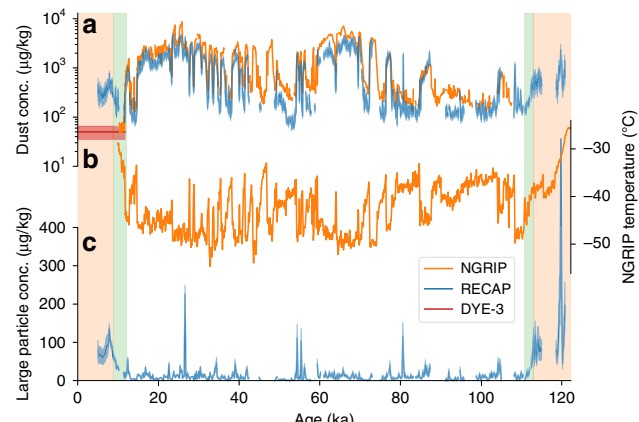

**Fig. 1** Dust and temperature records from Greenland ice cores. **a** The total concentration of dust particles in the size range 1.25–10.5 µm in the RECAP and NGRIP ice cores over a full glacial cycle on 50 year resolution. For DYE-3 the plot shows the average Holocene value. The coloured shadings around the curves are 1σ errors. **b** NGRIP temperature reconstructed from atmospheric nitrogen isotope ratios[21]. **c** The large (8.13–10.5 µm) particle concentration in the RECAP ice core on 200 year resolution. The background colours indicate periods of high (orange) and low (white) large particle concentrations and the transition between the two states (green), determined by a piecewise continuous rampfit function

Considering the glacial section (11.7–119.0 ka b2k) of the RECAP dust record, we find similar features of the dust record (concentrations, particle size mode, scales of variability) to central Greenland ice cores. RECAP dust concentrations varied by a factor of 10–100 between mild interstadial and cold stadial periods commonly known as Dansgaard-Oeschger events[18]. These changes are attributed to changes in aridity and dust storm activity in central Asian deserts as well as hemispheric-scale atmospheric circulation patterns[19,20]. We compare the RECAP dust record to the NGRIP ice core, which is the longest continuous central Greenland dust record available[10]. We observe a Pearson correlation coefficient of 0.95 ± 0.01 between the log-scaled 100-year mean RECAP and NGRIP glacial dust records. The NGRIP dust concentration is 1.7 ± 0.2 times greater than the RECAP dust concentration. Assuming an identical dust flux to the two sites, the different dust concentrations can be explained by the different snow accumulation rates at NGRIP (19 cm ice equivalent yr$^{-1}$)[21] and RECAP (45 cm ice equivalent yr$^{-1}$). The accumulation difference alone explains the higher dust concentrations in NGRIP compared to RECAP, without the need to invoke differences in source activity or atmospheric dust transport. The mode of the RECAP glacial dust size distribution, i.e. the particle size contributing most to the total mass (see Methods for calculation of the mode), is also in good agreement with those reported for central Greenland ice cores. The RECAP glacial dust size distribution mode is 2.22 ± 0.02 μm, compared to 1.73 μm for NGRIP (Fig. 2)[10]. The close similarities of dust concentrations and particle size distributions strongly suggest glacial dust deposited at RECAP, GRIP, GISP2 and NGRIP originated from a common source[18]. In the absence of geochemistry data for RECAP glacial dust, we assume that the central Asian dust source determined for other Greenland ice cores[12,13] also provided dust to Renland ice cap throughout the glacial.

The RECAP dust record also confirms previous findings of surprisingly high interglacial dust concentrations at coastal

Greenland ice core sites compared to central Greenland ice core sites. Bory et al.[14], reported elevated late Holocene dust concentrations in coastal Greenland ice cores from Renland (1360 μg kg$^{-1}$, dated 1604–1662 CE) and Hans Tausen (476 μg kg$^{-1}$, dated ca. 1000 CE) ice caps, which are similar to those reported here for RECAP interglacial samples. In comparison, dust concentrations in contemporaneous central Greenland ice core samples ranged from 35 to 123 μg kg$^{-1}$. The elevated interglacial dust concentrations found at Renland and Hans Tausen suggest an alternate or additional source of dust impacts on coastal East and Northeast Greenland but not the central Greenland ice core sites.

Further support for an additional coastal Greenland interglacial dust source is provided by particle size distributions in the RECAP ice core, which reveal a much larger particle mode than those found in central Greenland ice cores. The mode of the RECAP Holocene size distribution is 19.6 ± 1.0 μm, compared to 1.47 μm for NGRIP (Fig. 2). Furthermore, the NGRIP and RECAP glacial dust size distributions are concave with power law tails, whereas the Holocene RECAP size distribution is bimodal and therefore indicative of contributions from two distinct sources. A power law was fitted to the tail of the RECAP Holocene particle size distribution between 4 and 6 μm and the concentration of excess particles in the size range 0.9–2.5 μm was determined. The mode of these small RECAP Holocene particles is 1.88 ± 0.04 μm (Fig. 2, see Methods for details on separating the two distributions), consistent with that of NGRIP early Holocene ice. The large RECAP Holocene particle mode (19.6 ± 1.0 μm) suggests a dust source local to Renland ice cap, as such large particles are rapidly sedimented and typically reside in the atmosphere for less than a day[19].

To better characterize changes in large and small particle deposition in the RECAP dust record, we identify two particle size ranges representative of these modes and determine their concentrations over time. The small (1.25–2.9 μm) and large (8.13–10.5 μm) particle size ranges are shown in Fig. 2 and, respectively, correspond to size ranges assigned to 'distal' and 'local' dust sources in a comparable study[22]. The record of RECAP large particles (Fig. 1) shows elevated values during interglacials (Holocene 66 ± 28 μg kg$^{-1}$ and late Eemian 205 ± 170 μg kg$^{-1}$, durations as previously defined) and low values during the glacial (14 ± 21 μg kg$^{-1}$). During the Holocene, RECAP large particle concentrations peak (135 μg kg$^{-1}$) at 7.8 ± 0.1 ka b2k coinciding with the Holocene climatic optimum at Renland[23]. This coincidence may mark the maximum extent and/or activity of the local East Greenland dust production zone, which is the product of both the retreating ice sheet margin and the lowering relative sea level. Northern Hemisphere insolation was at a maximum during the Holocene climatic optimum, implying a maximum rate of meltwater runoff from the Greenland ice sheet even if the ice sheet continued to lose elevation (and therefore mass) until ~7 ka b2k[23]. Available reconstructions indicate that the central east Greenland deglacial response in relative sea level change was almost complete by 8 ka b2k[24–26]. This suggests that relative sea level decrease may have had a greater influence than ice margin retreat with respect to the establishment of dust production areas local to Renland ice cap. Three instances of high concentrations of large particles are found in the glacial, and all correspond to the ages of tephra layers previously identified in central Greenland ice cores (26, 55 and 81 ka b2k)[27], although geochemical measurements have not yet been undertaken to confirm a volcanic source for these large particles. Otherwise, RECAP large particle concentrations vary by a factor of just 2.6 ± 0.5 throughout the glacial stadials and interstadials (Supplementary Note 2, Supplementary Fig. 5), which is consistent with a factor 2 variability in the snow

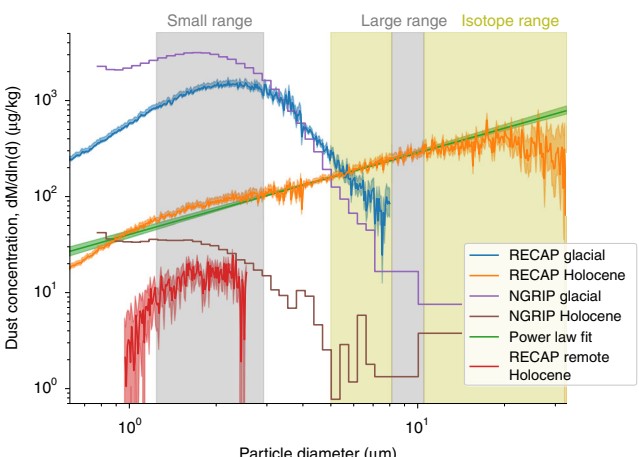

**Fig. 2** Dust size distributions in RECAP and NGRIP ice cores. RECAP dust size distributions were measured by Coulter Counter. The glacial data cover the time period (12,800–33,900) ± 500 years b2k, while the Holocene covers selected samples from the period (356 ± 2 − 4010 ± 50) years b2k. A power law is fitted to the Holocene data from 4 to 6 μm and extrapolated to 0.9 μm, with the residual Holocene dust relative to the power law fit shown in red. It is interpreted as the size distribution of the RECAP remote Holocene dust. The coloured shadings around the curves are 1σ errors. The grey bars show the size ranges assigned to represent small and large particles in the RECAP dust time series. The yellow area shows the size range of particles sampled for Sr and Nd isotope measurements

accumulation rate in central Greenland[28,29]. If the snow accumulation rate at Renland follows the same pattern, the glacial large particle concentration variations can be explained solely by variations in snow accumulation, suggesting the location of the central East Greenland ice sheet margin did not change significantly through the rapid stadial/interstadial cycles. Furthermore the data suggest a small but constant flux of dust from local sources was transported to Renland ice cap throughout the glacial.

**RECAP dust source apportionment.** The large particle size of the RECAP Holocene dust limits its atmospheric residence time to less than a day[30], thus we can evaluate Greenland and Iceland as the only possible sources of the RECAP interglacial dust. Although there are several publications dealing with Icelandic dust[31], few studies are available for Greenland, and these are limited to the Kangerlussuaq region in West Greenland[31,32]. Atmospheric models show extreme disagreement regarding the quantity of dust transported from Iceland to East Greenland[24,25]. Nonetheless we can exclude Iceland as a source of RECAP dust on the basis of ice core dust geochemistry[14] as discussed later in this section. With regard to the likelihood of West Greenland as a source of coarse dust particles to East Greenland, we first consider the lack of large dust particles observed in ice cores from central and South Greenland[9,33]. Only a direct transport route across the ice sheet would be consistent with the limited atmospheric lifetime of the large particles found in RECAP ice. A much longer transport path around Southern Greenland can also be excluded as it is inconsistent with synoptic-scale circulation patterns[34].

To better characterize the origin of large dust particles observed in Holocene RECAP ice, we have measured strontium and neodymium isotope ratios in RECAP ice samples as well as in exposed rock and sediment samples from central East Greenland (Kong Christian X Land and Scoresby Sund). Strontium and Neodymium isotope ratios were measured in particles larger than 5 μm for three RECAP samples corresponding to the period 4–7 ka b2k (Fig. 3). The $^{87}Sr/^{86}Sr$ ratios in RECAP Holocene ice (0.745–0.751) are consistent with values from Renland (0.739) and Hans Tausen (0.748) ice cores, which were attributed to proximal dust sources of Proterozoic and Paleozoic age[14]. As shown in Fig. 3, $^{87}Sr/^{86}Sr$ ratios of RECAP, Renland and Hans Tausen ice core dust are distinct from those of Icelandic rock (0.703)[35] as well as central Greenland ice cores (0.716–0.719)[14]. We have also measured $^{87}Sr/^{86}Sr$ ratios in dust samples collected from Scoresby Sund region (0.716–0.736) and exposed rock samples from Kong Christian X Land (0.747–0.772). The specific locations sampled in Kong Christian X Land, Skræntdal (72°34'N, 27°29'W) and Nunatakgletscher (73°55'N, 25°54'W), are respectively located 150 km and 300 km north of Renland ice cap[15]. The Sr and Nd isotope signatures are consistent with a Renland ice cap dust source located within Kong Christian X Land (Fig. 3). The source location may have varied in the past; the isotopic signature of RECAP ice from 5 to 6 ka b2k is consistent with a Nunatakgletscher-region source, but earlier (6-7 ka b2k) and later (4-5 ka b2k) samples are more consistent with a Skræntdal-region source.

Until a comprehensive sampling campaign is undertaken we attribute the RECAP large dust particle fraction to dust deflation sources within Kong Christian X Land located between 72 and 74°N. Such a source is consistent with observations of northerly wind flow along the central East Greenland coast[36]. Strontium and Nd isotopic signatures of RECAP samples are not consistent with dust deflation sources within Scoresby Sund, most likely because the strong katabatic winds in the fjord system prevent transport of coastal materials upslope to the ice core site. The Worldview satellite observed a dust storm rising from the glacier

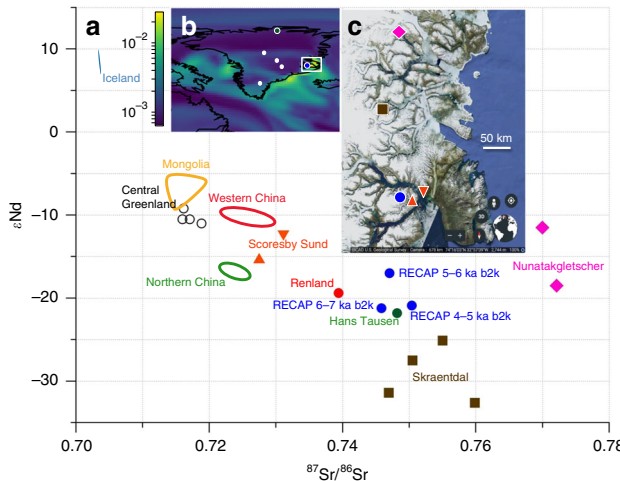

**Fig. 3** Sr/Nd isotopic signatures of dust in Greenland ice cores and potential dust source regions. **a** Strontium and neodymium isotope ratios from RECAP (blue circles) compared to possible dust source regions in Greenland (Scoresby Sund and Kong Christian X Land, this study), Iceland[35] and Asia; and ice core samples from Renland (red circle), Hans Tausen (green circle) and central Greenland (DYE-3, NGRIP, GRIP, Site A, empty circles)[14]. Errors are 2σ, but smaller than the data markers. The RECAP samples each cover approximately 1 ka. The Hans Tausen sample is from approximately 1000 CE while all other Greenland samples are from the 17th-18th century. The Renland ice core (red circle) was drilled in 1988 at a location less than 2 km from the RECAP ice core (blue circles). **b** Copernicus Atmosphere Monitoring Service reanalysis map of total aerosol optical depth from April 27 2016[37] indicating dust deflation zones along central East Greenland. The coloured circles indicate ice core drill sites; from north to south: Hans Tausen, NGRIP, GRIP, RECAP, Site A and DYE-3. The white box shows the area in subplot c including the RECAP ice core location (blue circle). Generated using Copernicus Atmosphere Monitoring Service information [2018]. Neither the European Commission nor ECMWF is responsible for any use that may be made of the Copernicus Information or Data it contains. **c** Locations of potential dust sources (triangles, square, diamond) proximal to RECAP ice core drilling site (blue circle). Google Earth map data: IBCAO US Geological Survey

outwash plain in Schuchert Dal less than 100 km from the RECAP drill site (Supplementary Fig. 6). Satellite reanalysis data (Fig. 3) show local dust plumes originating from the East Greenland coast and extending several hundred kilometres to the East[37]. We infer that outwash plains are the dominating dust source on the East Greenland coast, similar to the role these features play in Iceland and West Greenland. Therefore the interplay between relative sea level changes and ice sheet advance or retreat were key controllers of local dust availability through the last glacial cycle.

## Discussion

Drawing from the hypothesis that large dust particles observed in RECAP interglacial ice originate from local dust sources, we investigate the timing of decreases and increases in large particle concentration at the onset and termination of the last glacial period, respectively. To determine the timing, we fit a piecewise continuous function with two constant functions connected by a linear function to the ratio between large and small particles. We call it a rampfit, following Mudelsee[38]. The free parameters in the fit are the timing and concentration values at the kink points between the constant and linear functions, and their timing is interpreted as the onset and termination of the transition between glacial and interglacial or vice versa.

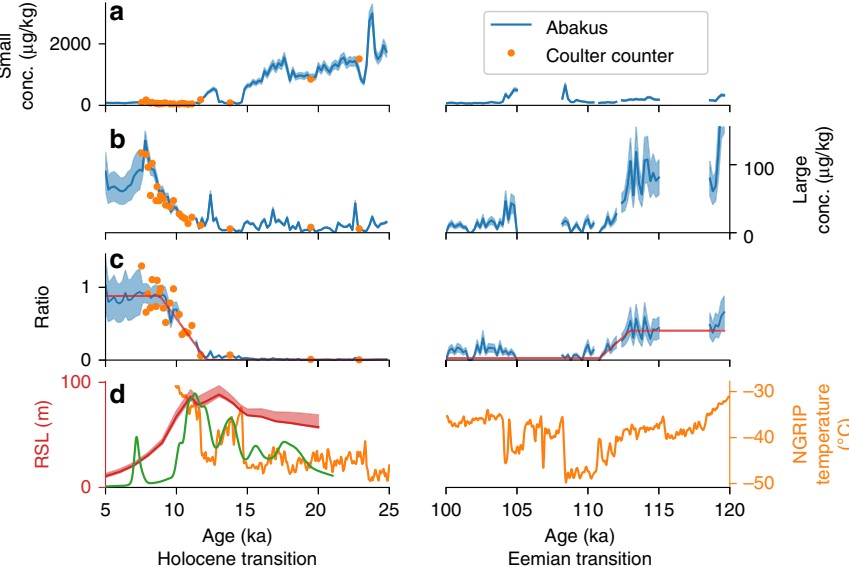

**Fig. 4** Glacial-interglacial transitions in RECAP dust. The small (1.25–2.9 μm) (**a**) and large (8.13–10.5 μm) (**b**) particle concentration and the ratio (**c**) between the large and small particle concentrations found in RECAP ice core are shown. 200 year means are shown for dust particle sizes measured by an Abakus laser particle sensor from Klotz GmbH, Germany (blue) together with 55 cm Coulter Counter samples (orange) and a piecewise continuous rampfit function applied to the Abakus data (red). The blue shadings around the curves are 1σ errors. **d** NGRIP temperature (orange), relative sea level in Scoresby Sund (red)[45] and probability density function (PDF) of deglacial [10]Be ages for the Scoresby Sund region[43] (green). The deglacial [10]Be PDF shows the number of rock exposure timing measurements in the Scoresby Sund region through the glacial termination

At the onset of the last glacial period, the East Greenland ice sheet margin advanced and relative sea level subsequently rose due to isostatic depression. The rampfit method applied to the RECAP dust record shows that most local dust sources were extinguished from 113.4 ± 0.4 to 111.0 ± 0.4 ka b2k (Fig. 4). The concentration of large dust particles is greater during the Eemian than during the Holocene, consistent with a smaller Greenland ice sheet[2,39] with ice sheet margins located further inland than at present. Given that isostatic bedrock depression can only occur subsequent to ice sheet growth, we infer that the extinction of dust sources in Kong Christian X Land occurred as a result of ice sheet margin advance and not as a result of increasing relative sea level, although improved modelling of glacial isostatic adjustment and contemporaneous changes in sea level are required to fully resolve this question.

Even though the large particle concentration dropped by more than 90% after the Eemian, the remaining 14 ± 21 (1σ) μg kg[−1] suggests that some local dust sources may have existed throughout the glacial. While almost all of East Greenland was covered by ice during the last glacial maximum[1,40], it is still disputed whether there were exposed ice-free areas present during the glacial in the form of low-lying ice-free land or exposed nunataks[1,41,42]. The presence of large particles throughout the glacial in the RECAP record is consistent with the existence of exposed, albeit small, local dust sources throughout that period. No consistent pattern of large particle concentration variability is observed between stadials and interstadials, suggesting that rapid climate oscillations did not have a significant impact on the activity of local dust sources.

For the deglacial transition, the rampfit analysis indicates an increase in local dust sources occurred from 12.1 ± 0.1 to 9.0 ± 0.1 ka b2k. This change is almost two orders of magnitude slower than the changes in geophysical proxies recorded in the NorthGRIP ice core during the deglaciation. The observed NorthGRIP ice core proxy changes were: reduced aridity of Asian deserts (inferred from dust particle concentrations with a mode of 1.7 μm), shifts to Northern Hemisphere atmospheric circulation

patterns (inferred from sodium concentrations), changes to precipitation moisture sources (inferred from deuterium excess) and increasing Greenland air temperatures (inferred from δ[18]O)[19]. We propose that the much slower increase in RECAP large particles over the deglaciation is due to dust source availability changes caused by retreat of the ice sheet margin and contemporaneous changes to relative sea level in Kong Christian X Land. We speculate that these changes led to the exposure of particle-rich glacial outwash plains, some of which may continue to serve as active dust sources even in the present day.

The timing of the deglacial increase in RECAP large particle concentrations is consistent with available evidence of ice margin retreat and relative sea level decrease in the region. Glacier retreat has been measured by radiocarbon dating of exposed organic material and cosmogenic surface-exposure dating of rocks[3,43]. Nunatakgletscher terminates into Kejser Franz Joseph Fjord, which was sequentially deglaciated from 15.3 to before 9.0 ka BP on the continental shelf and from before 9.0–7.4 ka BP in the fjord[44]. Radiocarbon-based deglaciation ages about 10.5 cal ka BP have been reported for coastal locations between Kejser Franz Joseph Fjord and Kong Oscar Fjord[40]. Coinciding rapid decreases in relative sea level of nearly 80 m have been documented in the nearby locations of Mesters Vig (10.5-8 ka BP), Hudson (10-9 ka BP) and Scoresby Sund (11-8 ka BP)[45]. A peak frequency of deglacial [10]Be ages is also found in Scoresby Sund around 11 ka b2k (Fig. 4). The steep valleys of the Kejser Franz Joseph and Kong Oscar Fjord systems ensure that low-lying flat areas such as outwash plains are sensitive to increases in relative sea level and may be rapidly inundated even if they are located far from the coastline.

The RECAP ice core dust particle size distribution is applied as an indicator of local dust sources, providing insights into the status of the central East Greenland ice sheet margin within the 72–74°N sector of Kong Christian X Land. High concentrations of large dust particles during the Eemian suggest ice sheet margins were located further inland than at present, supporting various lines of evidence indicating a smaller Greenland ice sheet

during the Eemian. At the glacial onset, the ice sheet margin advanced to cover local dust sources from $113.4 \pm 0.4$ to $111.0 \pm 0.4$ ka b2k, with a small flux of large particles to Renland ice cap through the glacial. The low but nonzero large dust particle concentration in the ice core record supports the possibility of ice-free land in central East Greenland throughout the glacial. The RECAP dust record shows that dust sources in Kong Christian X Land became exposed from $12.1 \pm 0.1$ to $9.0 \pm 0.1$ ka b2k, consistent with relative sea level estimates[45] and previous measurements of ice sheet retreat[43]. This dust record provides new constraints on the location of the Greenland ice sheet margin through the last glacial cycle, with potential implications for millennial-scale reconstructions of the Greenland ice sheet response to climate forcings.

## Methods

**Dust measurements by abakus laser particle counter**. The RECAP dust record was measured using an Abakus laser particle sensor (Klotz GmbH, Germany) connected to an ice core melting continuous flow analysis system[46]. The Abakus measures particle concentration as a function of size. The size bins are calibrated by comparing to Coulter Counter data[47], and cover the range 0.64–9.6 µm. The depth resolution of the Abakus measurements is 0.5 cm. With an annual snow accumulation rate of 45 cm ice equivalent, this gives sub-annual resolution down to 4 ka b2k. However, due to extreme ice sheet thinning the layer thickness diminishes down through the core, so in the glacial ice, 1 cm corresponds to 100 yr. The two categories small and large are used for particles in the range 1.25–2.9 µm and 8.13–10.5 µm, respectively. These ranges give the best separation between small and large particles, since the Abakus measures particles between 1.25 and 10.5 µm. All ages are measured relative to 2000 CE, and the conversion from depth to age follows the RECAP time scale (Supplementary Note 1). All ages of ice core samples are based upon the GICC05modelext time scale[18].

The mode of the particle size distributions is defined as the particle size corresponding to the maximum of the distribution. All modes calculated or mentioned here are modes of probability density functions for particle volume or equivalently for mass. The probability density function for particle mass, $\frac{dM}{d\ln(d)}$, is defined such that $\int_{\ln d_1}^{\ln d_2} \frac{dM}{d\ln(d)} d\ln(d)$ gives the total dust mass of particles in the diameter range $[d_1, d_2]$. To reduce the effect of measurement error on the mode, we measure the mode as the maximum of a parabola fitted to a size range of the distribution around the maximum. The fit interval is chosen so small that the distribution closely resembles a parabola, but so large that there are enough points to reduce the effect of measurement error. For the glacial this corresponds to the size range 1.4–3.7 µm and for the Holocene 12.3–27 µm. To separate the local and remote contributions to the Holocene size distributions, we assume that the size distribution of the local dust has power law tails like the glacial dust. Any additional dust is assumed to come from remote sources. The lower tail is determined by fitting a power law to the Holocene size distribution between 4 and 6 µm (green line in Fig. 2). The additional dust from 1 to 3 µm above the power law is interpreted as the dust from remote sources, and it is found by subtracting the power law from the RECAP Holocene distribution. The ratio between the remote Holocene dust concentration and the glacial dust concentration is calculated as a ratio between the maximum values of the RECAP glacial and RECAP remote Holocene distributions. The maximum concentrations are found similarly to the modes by fitting a parabola. For the glacial distribution the same fit interval as for the mode was used, while for the remote Holocene distribution the interval 1.17–2.04 µm was used.

**Abakus calibration**. To calibrate the Abakus data, parallel ice sticks 55 cm long were measured by Coulter Counter at the University of Milano-Biccocca Ice Lab (see Section 9 for data availability). The samples were decontaminated by washing the outer surfaces three times with ultrapure water. They were measured both with a 100 µm and a 30 µm aperture for accurate size determination of both large and small particles. The choice of samples was based on sample availability, and all measured samples have been included in this study.

**Geochemical isotope analysis of ice core samples**. Strontium and neodymium isotope analysis of RECAP ice was conducted at Lamont-Doherty Earth Observatory (LDEO). Melted ice samples were filtered through pre-cleaned 5 µm-pore size, 47 mm polycarbonate filters, which were then placed in pre-cleaned, MilliQ-filled Teflon 22 ml beakers. The beakers were sonicated for 15 min before the filters were removed and the filtrate evaporated. The recovered particles were digested using a 2:1 mixture of 7 N double-distilled nitric acid (produced at LDEO) and SeastarQR Ultrapure concentrated hydrofluoric acid. Closed Teflon beakers were sonicated for 15 min, then heated at 150 °C for 24–48 h or until complete digestion of the sample was observed. Ten percent of each sample was taken for concentration analyses.

Strontium and neodymium were isolated by column ion chromatography. The samples were initially passed through a 100 µl column of Eichrom AG1-X8QR 100–200 mesh anion exchange resin. From the recovered solution, REEs were isolated from the alkaline earth metals using 100 µl columns of Eichrom TRUQR resin (100–200mesh). Neodymium was then separated from the REE cut by elution through calibrated Eichrom LNQR resin using 0.25 N $HNO_3$ as eluent. Strontium was separated from the other alkaline earth metals via Eichrom SrQR resin, eluting with MQ water.

Isotope ratios were determined on a ThermoScientific Neptune Plus MC-ICP-MS, with an Apex desolvator for Nd analyses, and a Peltier chilled spray chamber for Sr analyses. Samples and standards were run at a concentration of 200 ppm. Mass-fractionation was corrected using the Rayleigh exponential mass-fractionation law assuming $^{86}Sr/^{88}Sr = 0.1194$ and $^{146}Nd/^{144}Nd = 0.7219$. All sample measurements were bracketed with NIST SRMs and standardised to the long-term, mass-fractionation-corrected average of the respective standard: NIST SRM 987 for Sr ($^{87}Sr/^{86}Sr = 0.710240$) and JNdi for Nd ($^{143}Nd/^{144}Nd = 0.512115$)[48]. We report $^{143}Nd/^{144}Nd$ ratios, and also further normalize those ratios to the Chondritic Uniform Reservoir (CHUR), where $\varepsilon Nd = 10^{4*}$ ($^{143}Nd/^{144}Nd_{Sample} - {}^{143}Nd/ {}^{144}Nd_{CHUR})/^{143}Nd/^{144}Nd_{CHUR}$ and $^{143}Nd/^{144}Nd_{CHUR} = 0.512638$. External reproducibility of the samples was checked against BCR2 (Sr) and La Jolla (Nd) standards. Our observed BCR2 value was $^{87}Sr/^{86}Sr = 0.705009$ ($2\sigma = 26$ ppm), within $2\sigma$ of the reference value 0.705000 reported by Jweda et al[49]. We found 0.511864 ($2\sigma = 20$ ppm) for La Jolla, also within $2\sigma$ of 0.511858 reported by Tanaka et al[48].

**Geochemical isotope analysis of dust source samples**. Strontium and Neodymium isotopic compositions have been determined in rock powder and sediment samples from central East Greenland, to characterize potential source areas of RECAP dust. Ten sediment samples were collected in the Scoresby Sund region (see Supplementary Note 3 for locations and data). For each sample, about 500 g of the upper 5–10 cm were taken. They were transported in sealed polyethylene bags and afterwards stored refrigerated. Six of the samples were analyzed for their strontium and neodymium isotope ratios. The samples were processed in two fractions each, a bulk sample and a sample that was sieved to 45 µm grain size. Six rock powder samples from Nunatakgletscher and Skræntdal were obtained from the Danish Geological Survey[15] (see Supplementary Note 3 for locations and data).

The samples were ground by hand in an agate pot with an agate pestle, and dissolved in a mixture of concentrated HF and aqua regia, in Savillex™ teflon beakers on a hotplate at 130 °C for 48 h. A mixed Sm-Nd spike was added during the dissolution process. Samples were repetitively ultrasonicated during this procedure to minimize the formation of calcium fluorides that usually have negative effects on the digestion of the samples. Sm–Nd and Sr isotopic compositions were analyzed using a VG Sector 54 IT Thermal Ionization Mass Spectrometer (TIMS) at the Department of Geoscience and natural Resource management, University of Copenhagen, Denmark. Strontium and REE were separated using chromatographic columns charged with 12 ml AG50W-X 8 (100–200 mesh) cation resin. Neodymium and Sm were further separated using smaller chromatographic columns containing Eichrom™ LN resin SPS (Part#LN-B25-S). A standardized 3 M $HNO_3$-$H_2O$ elution procedure applying self-made disposable mini-extraction columns, including 50–100 mesh SrSpec™(Eichrome Inc.) resin, was used for purification of Sr fractions. Samarium and Nd isotopes were collected in a static and dynamic multi-collection mode, respectively, using a triple Ta-Re-Ta filament assembly. Mass bias correction of the measured Nd isotope ratios was made using the $^{146}Nd/^{144}Nd$ ratio of 0.7219. The JNdi standard measurements yielded a mean value of $^{143}Nd/^{144}Nd = 0.512095 \pm 11$ ($2\sigma$; $n = 6$) during the period of analyses. Precision for $^{147}Sm/^{144}Nd$ ratios is better than 2% ($2\sigma$). Hundred ng loads of the NBS 987 Sr standard yielded $^{87}Sr/^{86}Sr = 0.710239 \pm 0.000016$ ($n = 7$, $2\sigma$). The $^{87}Sr/^{86}Sr$ values of the samples were corrected for the offset relative to the certified NIST SRM 987 value of 0.710247.

**Satellite images**. Figure 3 contains publicly available satellite data. The reanalysis data (Fig. 3b) are from the Copernicus Atmosphere Monitoring Service Near Real Time model[37]. We have used the 3 h prediction of the'Dust Aerosol Optical Depth at 550 nm' product. The data are from the $+ 3$ h prediction at April 27 2016, 00:00. The image of central East Greenland coast (Fig. 3c) is from Google Earth Pro V 7.3.2.5495, centered on 71°49'12"N 11°38'12"W with an eye altitude of 772 km. Copyright: The US Geological Survey.

**Statistics**. All errors are $1\sigma$ errors except for Sr and Nd isotopes, which are $2\sigma$. The Coulter Counter is assumed to have a 5% systematic error, a 2% random error following Delmonte et al.[50], and a $1/\sqrt{N}$ counting error. The Abakus has a 10% systematic error from the calibration to the Coulter Counter for samples measured both by Coulter Counter and Abakus. For samples where no Coulter Counter measurements were performed, an additional systematic error of 5% is added to the small particles and 30% to the large particles due to potential drift of the system, following Simonsen et al.[47]. The random error on the Abakus data is less than 5%, as we present 100 and 200 yr means, where the random error is suppressed by the large number of data points. All propagated errors are calculated using either linear

error propagation or bootstrap resampling following Mudelsee[38]. The errors on the start and end of the dust ratio transitions are based on the ramp fits without weighing the data points according to their uncertainty, as this would bias the result towards lower values.

## Data availability

All data generated in this study are available for download at www.iceandclimate.dk/data as well as NOAA Paleoclimate and PANGAEA (https://doi.org/10.1594/PANGAEA.906114) databases.

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

## Acknowledgements

The RECAP ice coring effort was financed by the Danish Research Council through a Sapere Aude grant, the NSF through the Division of Polar Programs, the Alfred Wegener Institute, and the European Research Council under the European Community's Seventh Framework Programme (FP7/2007-2013)/ERC grant agreement 610055 through the

Ice2Ice project. This is TiPES contribution 2: This project has received funding from the European Union's Horizon 2020 research and innovation programme under grant agreement No 820970. The Centre for Ice and Climate is funded by the Danish National Research Foundation. T.A.S. acknowledges NSF support for this work under two grants (1443464 and 1545162).

## Author contributions

A.S., H.A.K., M.F.S. and P.V. measured the Abakus data. B.D., G.B. and M.F.S. measured the Coulter Counter data. Strontium and neodymium isotope measurements were carried out by A.B., G.W., M.F.S. and S.G. (ice samples) and R.F. (sediment samples). D.V. and T.B. measured $CH_4$ and T.S. measured $\delta^{18}O_{atm}$ for the time scale. A.G. processed the satellite images. M.W. layer counted the upper 458.3 m of the ice core. B.V., D.V., M.W., T.B. and T.S. made the time scale. All authors contributed to interpreting the data and writing the manuscript.

## Competing interests

The authors declare no competing interests.
