## [Peer Review File · Nature Communications]

Reviewers' comments:

Reviewer #1 (Remarks to the Author):

Review of "Ice core dust particle size reveals past glacier extent in East Greenland" by Simonsen et al.

Summary

In this paper the authors use data about dust particle size distribution from an ice core to constrain the timing of local glacier advance and retreat. This adds an independent time constraint on glacier advance and retreat. They provide convincing arguments for their interpretation of the data.

There isn't much direct discussion about how localized this type of data is, although partially mentioned throughout the paper in bits.

One thing that could be clearer is the power law fit, shown in Figure 3. Just a few words where first mentioned would probably do the trick.

The paper (manuscript) feels a bit like it could do with a careful finishing touch, especially the last few appendixes.

Title - sounds like a single particle can be used.

Perhaps "Ice core dust particle sizes reveal past glacier extent in East Greenland"

Abstract - good (tiny suggestions).

"Here we USE [apply - out] a continuous ..."

"Strontium/Neodymium isotope ratios OF THOSE SMALLER PARTICLES are ..."

Missing "Introduction" section?

"Considering the NGRIP and RECAP accumulation rates (19 [14] and 45 cm ice equivalent per year, respectively), dry deposition could explain ..."

- change to "... rates, 19 [14] and 45 cm ice equivalent per year, respectively, dry ..."

Figure 1. Caption for c) and y-label don't seem to agree, caption "c: The ratio between ..." and y-label y-label says "Large particle conc. ($\mu\text{g}/\text{kg}$)"

Explain how you get the size ranges, small (1.25-2.9 μm) and large (8.13-10.5 μm).

Only mentioned briefly in appendix now.

p7. How is the the large mode of the RECAP Holocene dust distribution (19.6 \pm 1.0 μm) obtained?

Section 2

p7. Reference to Figure 3 earlier, and then Figure 5 here. Should be Figure 4.

p8. Figure references don't make much sense here. Figure 3 not showing isotope ratios, ...

Section 3

Past glacier extent of the Scoresby Sund region

 shouldn't it be "past glacier extent IN the ..."?

Figure 4. Description of arrows confusing.

p11. "To separate ..." could be explained a little more.

Don't have background to know whether Ne-Strontium method is good or not.

p12. Figure 5 description referring to some other figure. Also comes a little out of the blue there.

Reference list.

Is it the format to have et al. in the reference list?

Figure 6, perhaps not so easy to see tie point features.

p23. Figure numbers seem to be all over (wasn't the satellite at some point called Figure 10).

Is "Sample ..." a table? This seems all over the place here towards the end.

Sincerely, Throstur.

Reviewer #2 (Remarks to the Author):

This is an exciting manuscript that reports on the dust record from a small ice cap in East Greenland. The finding that the concentration of large dust particles increases during interglacial periods is interesting but perhaps not unexpected from this relatively low elevation, coastal ice core site. The most exciting results that come out of this dataset are the timing and magnitude of large dust particle concentrations during the Eemian and the Holocene. These are valuable results for interpreting past glacier extents, however they don't necessarily provide information that is more precise than the existing records of glacier change (at least for the Holocene). I also question whether the concentration of large dust particles tracks past glacier extent or, instead, relative sea level. I understand that the two are intimately related, but I bet that more surface area for dust is available as large outwash plains are exposed from a marine environment than from the recession of glaciers in terrestrial settings. I have some comments on the interpretation of an ice-free Jameson Land during the last glacial period (see below), but I wonder if the authors considered doing isotopic measurements to examine possible sources of large particle dust during the last glacial period? This might be useful for determining whether Jameson Land was a possible source. I find the manuscript itself difficult to read/understand even though the results are relatively straight forward. This and my lack of surprise about the results makes me less willing to support it for publication in Nature Communications. Below I highlight some of my concerns about the manuscript and suggest some changes. Some of these are minor editorial comments, but some raise more significant issues with the manuscript.

Page 2, lines 2-3. Need to clarify which age (i.e., Eemian, last glacial period, Holocene) dust the sentence refers to. Strontium/Neodymium isotope ratios of what?

Page 3, line 2. Less radiogenic than what?

Page 3, line 18. The concept of large dust particles hasn't yet been introduced in the manuscript (except in the abstract) and sort of comes out of the blue here. Suggest to first state what this is and then discuss its significance.

Page 3, line 36. Explain specifically how dry deposition influenced the difference in dust concentrations between RECAP and NGRIP.

Figure 1 caption. What are "ramp fits results"? What the "ramp fit" is and how it is determined need to be explained in the text (not the methods) since so much of the interpretations are based upon these results.

Page 5, line 1. What is the "volume mode"? This is used multiple times and should be defined at the first use.

Page 5, line 14. Insert "Holocene" between "reported elevated" and "dust concentrations".

Page 5, lines 24-26. What is the minimum Holocene Greenland ice sheet extent based upon? I

understand the authors are citing published work, but they should explain whether this work is data based or model based. Since the cited paper is a modeling paper that runs multiple experiments with varying results for Holocene Greenland ice sheet extents, the authors should clarify which experiment they are referring to and why they decided this was the most appropriate for comparison.

Page 5, line 34. What is the "small mode"? This and other similar terms need to be defined before using.

Figure 2 caption. Why are the samples used to characterize the Holocene dust size distribution only from 356-4010? And what is meant by "selected samples? How many samples and how were these selected? This makes me concerned that I don't fully understand the data used to come up with the results presented, and therefore that I can't critically assess the results or interpretations. The authors need to be very transparent about the samples and methods used to come up with the presented results.

Also, I would think one would want to look samples from the whole Holocene. Why are the samples only from the late Holocene?

Figure 2 caption, third to last sentence. What is meant by the "shaded areas"? Shaded areas on which graph?

Figure 3. If the interesting results (at least for the Holocene) are ~9-11 ka, I don't understand why the plot shows 5-25 ka. It would be great to have a larger/blown up section that focuses in on the transition/interpreted time of ice recession during the Holocene.

Figure 3 caption. Need to define what "Abakus" is in this text. I understand it is in the methods, but it should be clear to the reader here.

Figure 3 caption. Need to indicate the color of the line in the sentence "glacier retreat index for the Scoresby Sund region based on ^{10}Be measurements (blue)". Also the citation for this "glacier retreat index" seems to be wrong. The citation (#28) is Bory et al., 2002 and has nothing to do with ^{10}Be ages of glacier retreat. Also see my comment/question below about the "glacier retreat index".

Page 8, line 1. State specifically how the patterns are different for stadial-interstadial changes.

Page 8, line 6. Again, need to state specifically how dry deposition influences the results.

Page 8, section 2. In general, I find this section poorly organized. It would be helpful to start with a regional picture of atmospheric circulation in east Greenland and where one might expect local/regional sources of dust to occur that may have been transported to RECAP. Then, go into what the isotopic data support or don't support. As written, it seems like the authors exclude or include locations before explaining why they may/may not be possible sources.

Page 8, line 22. For example, here I'm expecting some explanation of the atmospheric circulation that influences the RECAP site. But that isn't discussed until later on.

Page 8, line 32. Need to state here specifically what isotopic composition is observed. Again here, the authors seem to get ahead of themselves by describing what is compatible/incompatible before simply saying what the results show.

Page 9, line 26. State specifically which results are used to make this interpretation.

Page 9, line 28. What is meant by the East Greenlandic cryosphere? What is this statement based upon? Seems unjustified by the data presented here.

Page 9, lines 32-34. This statement is not justified by the results presented here. Suggest to delete.

Figure 4. This figure needs a more informative caption, or perhaps better labeling/information in the figure itself. For example, what are RECAP 1, 2, and 3? I see that later on in the caption these are different age samples, but this should be explained clearly on the figure itself. Why are the isotopic values for these samples so different? What is Site A? Write out the meaning of "CAMS".

Page 11, lines 9-13. I found this point one of the most interesting of the paper. Is there a way to use this rate of change to investigate with more depth the differences between the glacial-interglacial/stadial-interstadial changes?

Page 11, line 17. What is the "glacier retreat index"? I understand what this is conceptually but I have no idea where it comes from. I searched the cited Sinclair et al. (2016) paper for the term and cannot find it there. If it is something the authors came up with their own index of glacier retreat based on ^{10}Be ages from the Scoresby Sund region that needs to be described in detail. I am not aware of reported glacier retreat in the Scoresby Sund region that starts increasing at 20 ka.

Page 11, lines 18-20. This statement just isn't true. As a glacier recedes, its surface lowers exposing valley walls at a similar time as its terminus retreats exposing outwash.

Page 11, lines 23-24. What is meant by the "relatively large statistical uncertainty of the large particle concentrations"? This is the first mention of a large uncertainty in the data. If this is an issue it should be discussed clearly in the first reporting of the results and at that point limitations should be set on interpretations.

Page 11, lines 26-27. What is meant by "a subsequent readvance"? What is this based on?

Page 11, lines 29-32. I think this is the big story here. It's the exposure of the large outwash plains with relative sea level fall. This is interesting and I think would be better to focus on this rather than other less well supported points above.

Page 12, line 1. What is the isotopic signature of the large particle dust during the last glacial period? This would be interesting to determine as a test of whether the particles originated from Jameson Land.

Page 12, lines 7-10. This is an interesting question about Jameson Land. I think it is reasonable to say that the large particle dust concentration during the last glacial period leaves open the possibility that Jameson Land was ice free, but it certainly doesn't prove it. Again, it would be interesting to do the isotopic signature of this dust to test the source.

If it is from Jameson Land, I'm surprised it doesn't change during stadial-interstadial changes because I would expect that region to be frozen/covered with snow during stadials.

Also, do the possible source areas change for large particle dust during the last glacial period (as compared to, for example, the Holocene) since winds were likely much stronger during the glacial period?

Page 12, line 11. This is an awkward statement here. It seems like "Here we propose a..." should be in an abstract or at least in the beginning of a manuscript, not in the final paragraph. Also, I don't necessarily agree that it is a "local paleo glacier extent" proxy. I would suggest it is a relative sea level proxy.

Reviewer #3 (Remarks to the Author):

This is an interesting and clearly written manuscript. It claims to demonstrate that large dust particles in the RECAP glacial ice originate from local glacier outwash plains in East Greenland. This is novel and finding the dust source is certainly of broad interest to the scientific community. The source, transport and deposition of fine dust on GrIS is relatively well understood whereas less is

known about the source of large particles. Clearly, their size does not allow long transport or residence time in the atmosphere, so a local/regional source is likely. It makes fully sense to look for a source in Greenland. I have some concerns that must be addressed before I could potentially be convinced completely about this new proxy for local glacier extent in Greenland. They are:

P. 2: "During the last glacial period, the ice core dust concentration was 10-100 times greater than in the Holocene due to enhanced continental aridity, increased wind strength and longer atmospheric particle lifetime [7-9]."

Are there any studies or data that show what effect the exposed continental shelves and shallow seas had as dust sources? Including the shelves around Greenland.

P. 3: "It furthermore shows that outwash plains became inactive 11.0-0.4 ka b2k during the post Eemian ice sheet advance."

Why did the outwash plains not become larger and more active when the glaciers advanced? At this time sea level was dropping so more fjords and continental shelf was being exposed (and could act as dust sources). The outwash plains can build out in the fjords as the sea level drops.

P. 5, Figure 2: Why do the glacial data only cover the period from 12,800-33,900 y b2k? And why do the Holocene data only cover the period 356-4,010 y b2k? Are there data for the whole glacial and the whole Holocene? And why exactly these periods? What if other periods were selected or if the complete glacial and Holocene were studied? The results would be much more convincing if they included the whole periods.

P. 7: "The relatively constant large particle flux to RECAP over stadial/interstadial cycles suggests that the activity of such local dust sources is not greatly affected from stadials to interstadials.

OK, very good. The data are convincing regarding this.

P.10: "However, exposure dates are typically measured on the valley sides which were exposed before the outwash plain [29]."

No. And this is not generally the case for CN datings such as the wording suggests. The ages from Kelly et al. (2008) are both from moraines on the valley floor and from moraines along the valley sides. The glacier front must be expected to retreat in pace with glacier thinning, so I would not expect the valley sides to be exposed prior to the frontal retreat (and, thus, increase of glacier forefield area).

P. 10: "with the warmest period of the Holocene Climatic Optimum."

Add reference documenting that HCO occurred at 8 ka at this site.

P. 10: "Exposure of outwash plains was further enhanced by decreasing relative sea level from 80 ± 10 to 40 ± 5 m above the present level from 11.0 to 9.0 ka b2k [31]. As most of Schuchert Dal is less than 50 m above sea level, the relative sea level lowering during the deglaciation would have greatly influenced the area of the outwash plain."

The opposite argument must then also be valid – i.e. when RSL lowers at the onset of the last glacial period (after the Eemian), the area of outwash plains increases.

P. 10: "lack of glacial erosion younger than the previous glacial (Saalian) suggests that it was ice free [1,33,34]."

Could as well be explained by a cover of thin, cold-based ice. Why do you think that is not a likely scenario?

P. 10: "points to the existence of exposed land despite glacial coverage of major dust sources."

What about the continental shelf E of Greenland? In particular the inter-trough areas.

P. 22: "Satellite images give direct evidence of dust storms in the Scoresby Sund region (Figure 10)."

The image does not show a dust storm, it is a view of a sediment-laden glacial meltwater plume flowing into the fjord on a good weather day. Icebergs can be observed in the fjord. Sediment plumes are seen in the water, not in the air! This is commonly observed at such deltas. Does the meteorological data support that there was a storm on August 12, 2012? As far as I can see, the Scoresbysund weather station was out of order that day but Danmarkshavn and Tasiilaq measured high pressure and slow winds. That does not exclude a local storm on this day in Schuchert Dal, but please check up on this.

The ice core analyses, data and statistical treatment appear very solid and sound. The reference to ice core literature is good, but the paper is relatively weak when it comes to linking it to surface processes and glacial geomorphology. An example is the use of "outwash plain". Not all land that becomes exposed by glacier retreat is an outwash plain. Much of it would typically be till plains (moraines) consisting of unsorted sediments (diamict) less prone to erosion by the wind than

outwash plains. Similarly, the land that becomes exposed by RSL lowering (regression) is a completely different surface than the outwash plains characterized by braided rivers. In conclusion, I think the dataset is very interesting and it could potentially be published here if ALL the issues are addressed. This could mean new data collection and analyses. I am also not convinced that the source of the coarse-grained dust is the outwash plains. At least not based on the attempted modern analogue of a "dust storm" from the satellite image (Fig. 10). I would like to see measurements from sediment traps or snow on the ice cap that can be clearly linked to wind erosion during observed storms of outwash plains near the Renland Ice Cap. This should be possible, and the phenomenon is well known from Iceland where the albedo of the ice caps can be dramatically changed during summers where dust storms erode the outwash plains and other loose sediment surfaces. This is most likely documented on satellite imagery (in E Greenland, too). Are there other potential coarse-grained sources in the landscape? Are there any high-altitude plateau in the mountains that might act as a source area?

Minor issues:

- P.1. "past extent of Greenland...", not paleo. This appears several places in the text.
- P. 2. "sea shells on raised beaches". Should be "marine mollusc shells" and note that one would prefer non-marine material for such analyses, e.g. driftwood. See e.g., Funder et al. (2011) in Science.
- P. 4. "stable water isotopes". Should be either stable oxygen or hydrogen isotopes.
- P. 6. In the caption to Fig. 3, add "(green)" to an explanation of the green curve in sub-figure d.
- P. 8. "Glacial and Holocene dust found in Central Greenland ice cores have been consistently attributed to Central Asian dust sources [12, 23]." This is a repetition – delete.
- P. 23. "Figure 9" should be "Figure 10".

Reviewers' comments:

Reviewer #1 (Remarks to the Author):

Dear Reviewer #1. Thank you for the constructive comments. We agree with all corrections and have tried to address them as best we can.

Review of "Ice core dust particle size reveals past glacier extent in East Greenland" by Simonsen et al.

Summary

In this paper the authors use data about dust particle size distribution from an ice core to constrain the timing of local glacier advance and retreat. This adds an independent time constraint on glacier advance and retreat. They provide convincing arguments for their interpretation of the data.

There isn't much direct discussion about how localized this type of data is, although partially mentioned throughout the paper in bits.

One thing that could be clearer is the power law fit, shown in Figure 3. Just a few words where first mentioned would probably do the trick.

OK. We have described ramp fit more clearly at the beginning of Section 4 (page 12, lines 18-24).

The paper (manuscript) feels a bit like it could do with a careful finishing touch, especially the last few appendixes.

Changes have been made throughout the manuscript in response to the three reviewers' comments, as well as to improve the flow and readability. These changes include revising the layout and expanding the caption for table 1.

Title - sounds like a single particle can be used.

Perhaps "Ice core dust particle sizes reveal past glacier extent in East Greenland"

Yes, changed.

Abstract - good (tiny suggestions).

"Here we USE [apply - out] a continuous ..."

"Strontium/Neodymium isotope ratios OF THOSE SMALLER PARTICLES are ..."

Yes, changed.

Missing "Introduction" section?

OK, added.

"Considering the NGRIP and RECAP accumulation rates (19 [14] and 45 cm ice equivalent per year, respectively), dry deposition could explain ..."

- change to "... rates, 19 [14] and 45 cm ice equivalent per year, respectively, dry ..."

Yes, changed.

Figure 1. Caption for c) and y-label don't seem to agree, caption "c: The ratio between ..." and y-label y-label says "Large particle conc. ($\mu\text{g}/\text{kg}$)"

Thanks - this has been corrected now.

Explain how you get the size ranges, small (1.25-2.9 μm) and large (8.13-10.5 μm).
Only mentioned briefly in appendix now.

The definition of the small and large size ranges is described in the Methods (page 14, line 3 to page 15, line 18):

The two categories "small" and "large" are used for particles in the range 1.25-2.9 μm and 8.13-10.5 μm . The exact limits of each range are defined by the size bins of the Abakus laser particle counter used. The Abakus measures particles in the range 1.25-10.5 μm .

The primary motivation for defining two size ranges is to differentiate between large particles with a short atmospheric lifetime (and therefore of local origin); and small particles with a long atmospheric lifetime and therefore distal origin. The small particle interval corresponds to a theoretical atmospheric lifetime of more than 1 week, whereas the large particle interval corresponds to an atmospheric lifetime of around 2 days.

p7. How is the the large mode of the RECAP Holocene dust distribution (19.6 +/- 1.0 μm) obtained?

This is now described in the Methods section (page 15, lines 1-18).

Section 2

p7. Reference to Figure 3 earlier, and then Figure 5 here. Should be Figure 4.

This has been corrected.

p8. Figure references don't make much sense here. Figure 3 not showing isotope ratios, ...

This has been corrected. It is now changed to Figure 4.

Section 3

Past glacier extent of the Scoresby Sund region

 shouldn't it be "past glacier extent IN the ..."?

Yes, it should. Corrected.

Figure 4. Description of arrows confusing.

Yes, changed to "upper" and "lower".

p11. "To seperate ..." could be explained a little more.

Yes, it has been expanded.

Don't have background to know whether Ne-Strontium method is good or not.

p12. Figure 5 description referring to some other figure. Also comes a little out of the blue there.

Yes, changed to Figure 4 b.

Reference list.

Is it the format to have et al. in the reference list?

Yes, that is the Nature Comms. format.

Figure 6, perhaps not so easy to see tie point features.

The methane tie point is defined as the maximum of the methane curve between 113 and 114 ka BP. A full list of tie points and their defining features are included in the associated data sets that will be released to public archives on acceptance.

p23. Figure numbers seem to be all over (wasn't the satellite at some point called Figure 10).

Fixed.

Is "Sample ..." a table? This seems all over the place here towards the end.

Sorry, the figures break the text inappropriately.

Sincerely, Throstur.

Reviewer #2 (Remarks to the Author):

This is an exciting manuscript that reports on the dust record from a small ice cap in East Greenland. The finding that the concentration of large dust particles increases during interglacial periods is interesting but perhaps not unexpected from this relatively low elevation, coastal ice core site. The most exciting results that come out of this dataset are the timing and magnitude of large dust particle concentrations during the Eemian and the Holocene. These are valuable results for interpreting past glacier extents, however they don't necessarily provide information that is more precise than the existing records of glacier change (at least for the Holocene). I also question whether the concentration of large dust particles tracks past glacier extent or, instead, relative sea level. I understand that the two are intimately related, but I bet that more surface area for dust is available as large outwash plains are exposed from a marine environment than from the recession of glaciers in terrestrial settings. I have some comments on the interpretation of an ice-free Jameson Land during the last glacial period (see below), but I wonder if the authors considered doing isotopic measurements to examine possible sources of large particle dust during the last glacial period?

We thank the reviewer for their interest in the work and for providing us with thoughtful and constructive comments.

Firstly we consider it important to note that Renland is not a particularly low elevation site: the drill site was at 2315 m asl, despite its geographical proximity to the coast. This is significantly higher than Camp Century (1887 m) and comparable to Dye-3 (2490 m) and NEEM (2450 m).

We have thoroughly revised our interpretation of potential local dust sources to Renland, including a consideration of impacts of relative sea level change. Further we have returned to Scoresby Sound to sample low-altitude areas and outwash plains and then conducted Sr-Nd isotope measurements. These results have been incorporated into the revised manuscript (with the addition of relevant coauthors) and are described further below.

Regarding the measurement of large particle dust during the glacial, this is unfortunately not possible due to the extremely low concentrations present in the ice. There is not enough sample available to conduct such measurements.

This might be useful for determining whether Jameson Land was a possible source. I find the manuscript itself difficult to read/understand even though the results are relatively straight forward. This and my lack of surprise about the results makes me less willing to support it for publication in Nature Communications. Below I highlight some of my concerns about the manuscript and suggest some changes. Some of these are minor editorial comments, but some raise more significant issues with the manuscript.

Page 2, lines 2-3. Need to clarify which age (i.e., Eemian, last glacial period, Holocene) dust the sentence refers to. Strontium/Neodymium isotope ratios of what?

Yes, they are Holocene, and it is strontium/neodymium isotope measurements of the dust. This has been added.

Page 3, line 2. Less radiogenic than what?

Less radiogenic than Central Greenland ice core dust. This has been added.

Page 3, line 18. The concept of large dust particles hasn't yet been introduced in the manuscript (except in the abstract) and sort of comes out of the blue here. Suggest to first state what this is and then discuss its significance.

Yes, we have added that we mean particles larger than 8 microns. We've added a line on particle sizes at the end of the second paragraph of the introduction,

Page 3, line 36. Explain specifically how dry deposition influenced the difference in dust concentrations between RECAP and NGRIP.

If the dust flux is the same at the two sites, and the dust is dry deposited, the concentration is inversely proportional to the snow accumulation. We have expanded on this in the paper (page 4, lines 11-14).

Figure 1 caption. What are "ramp fits results"? What the "ramp fit" is and how it is determined need to be explained in the text (not the methods) since so much of the interpretations are based upon these results.

Yes. The ramp fit is now explained in Section 4, and the description in the caption of Figure 1 has been changed.

Page 5, line 1. What is the "volume mode"? This is used multiple times and should be defined at the first use.

Yes. The mode is the particle size contributing most to the total mass. This has been added here and expanded upon in the methods.

Page 5, line 14. Insert “Holocene” between “reported elevated” and “dust concentrations”.
OK, added.

Page 5, lines 24-26. What is the minimum Holocene Greenland ice sheet extent based upon? I understand the authors are citing published work, but they should explain whether this work is data based or model based. Since the cited paper is a modeling paper that runs multiple experiments with varying results for Holocene Greenland ice sheet extents, the authors should clarify which experiment they are referring to and why they decided this was the most appropriate for comparison.
Yes, the modelling results are not as conclusive as we initially thought. We now compare the peak to the Holocene climatic optimum determined by Vinther et al. (2009).

Page 5, line 34. What is the “small mode”? This and other similar terms need to be defined before using.
Yes, it has now been defined in the “Dust record” section, paragraph 2.

Figure 2 caption. Why are the samples used to characterize the Holocene dust size distribution only from 356-4010? And what is meant by “selected samples? How many samples and how were these selected? This makes me concerned that I don’t fully understand the data used to come up with the results presented, and therefore that I can’t critically assess the results or interpretations. The authors need to be very transparent about the samples and methods used to come up with the presented results.
The samples were originally measured as part of a preliminary study before we started this present work (published in Simonsen et al., Clim. Past, 2018). They were chosen based on sample availability, and no measured samples have been omitted for this manuscript. The samples can all be seen in the Appendix E “Coulter Counter ages”, and a reference to this appendix has been added to the caption of Figure 2. We have added two lines on the choice of samples to the Methods.

Also, I would think one would want to look samples from the whole Holocene. Why are the samples only from the late Holocene?

We agree with the reviewer that it would be ideal to sample the whole Holocene, but both sample availability and laboratory/instrument time precluded this. We note that previous publications of dust particle size distributions in Greenland ice have found no significant variability in particle size mode between ice core locations (GRIP, NGRIP) and only a minor shift from glacial to interglacial (eg Steffensen et al., JGR 1997; Ruth et al. JGR, 2003). Thus we felt that the measurements presented are reasonably representative of the Holocene dust size distribution at RECAP. We have compared particle size distributions at different time periods of the Holocene and glacial. These are shown in Figure A at the end of this document. This, together with the flat large/small particle ratio in the early Holocene (Figure 3), suggests that the size distribution is the same over the Holocene.

Figure 2 caption, third to last sentence. What is meant by the “shaded areas”? Shaded areas on which graph?

The shaded areas around the size distributions. This has been added.

Figure 3. If the interesting results (at least for the Holocene) are ~9-11 ka, I don’t understand why the plot shows 5-25 ka. It would be great to have a larger/blown up section that focuses in on the transition/interpreted time of ice recession during the Holocene.

It is interesting to see the constant large/small particle ratios before and after the transitions and how the NGRIP temperature drops after the Eemian. We are concerned that the reader might lose sight of the big picture if we zoom in too much (see Figure B at the end of this document for an example). Finally we wanted to be consistent and cover the same time duration in each column.

Figure 3 caption. Need to define what “Abakus” is in this text. I understand it is in the methods, but it should be clear to the reader here.

OK, we have added “Abakus laser particle sensor from Klotz GmbH, Germany”.

Figure 3 caption. Need to indicate the color of the line in the sentence “glacier retreat index for the Scoresby Sund region based on 10Be measurements (blue)”. Also the citation for this “glacier retreat index” seems to be wrong. The citation (#28) is Bory et al., 2002 and has nothing to do with 10Be ages of glacier retreat. Also see my comment/question below about the “glacier retreat index”.

The color “green” has been added, and the citation is correct now.

Page 8, line 1. State specifically how the patterns are different for stadial-interstadial changes.
Yes, we have now written “ and the stadial/interstadial variability can be explained by the variability in snow accumulation.”

Page 8, line 6. Again, need to state specifically how dry deposition influences the results.
Yes, this has been explained and rephrased.

Page 8, section 2. In general, I find this section poorly organized. It would be helpful to start with a regional picture of atmospheric circulation in east Greenland and where one might expect local/regional sources of dust to occur that may have been transported to RECAP. Then, go into what the isotopic data support or don't support. As written, it seems like the authors exclude or include locations before explaining why they may/may not be possible sources.
Thanks, this has been reorganised so circulation is discussed before isotopes.

Page 8, line 22. For example, here I'm expecting some explanation of the atmospheric circulation that influences the RECAP site. But that isn't discussed until later on.
We have reorganised the section as suggested.

Page 8, line 32. Need to state here specifically what isotopic composition is observed. Again here, the authors seem to get ahead of themselves by describing what is compatible/incompatible before simply saying what the results show.
Yes, we have stated the results now.

Page 9, line 26. State specifically which results are used to make this interpretation.
We agree that this statement is unjustified and should be deleted. Furthermore, this section has been restructured following the reviewers' comments. The next two comments are also similarly influenced.
Page 9, line 28. What is meant by the East Greenlandic cryosphere? What is this statement based upon?
Seems unjustified by the data presented here.
We agree and have deleted the statement.

Page 9, lines 32-34. This statement is not justified by the results presented here. Suggest to delete.
We agree and have deleted the statement.

Figure 4. This figure needs a more informative caption, or perhaps better labeling/information in the figure itself. For example, what are RECAP 1, 2, and 3? I see that later on in the caption these are different age samples, but this should be explained clearly on the figure itself. Why are the isotopic values for these samples so different? What is Site A? Write out the meaning of “CAMS”.
The caption and figure have been updated and expanded. Site A is the name of an ice core measured by Bory et al (2003), comparable to DYE-3, NGRIP and GRIP (see Alley and Koci 1988) and the reference to Bory et al. has been included in the caption. CAMS has been written in full.

Page 11, lines 9-13. I found this point one of the most interesting of the paper. Is there a way to use this rate of change to investigate with more depth the differences between the glacial-interglacial/stadial-interstadial changes?

Yes, we have expanded on this (see page 12, lines 31-36): “The fast stadial-interstadial transitions in small particle concentration are seen throughout the glacial in both the NGRIP (Ruth et al., JGR 2003) and RECAP ice core. They are due to changing atmospheric circulation patterns (Schupbach 2018 Nature Comms). We suggest that the much slower glacial-interglacial transitions in RECAP large particles are due to changes in source location and/or size caused by changes in glacier extent and relative sea level in the Scoresby Sund region.”

Page 11, line 17. What is the “glacier retreat index”? I understand what this is conceptually but I have no idea where it comes from. I searched the cited Sinclair et al. (2016) paper for the term and cannot find it there. If it is something the authors came up with their own index of glacier retreat based on ^{10}Be ages from the Scoresby Sund region that needs to be described in detail. I am not aware of reported glacier retreat in the Scoresby Sund region that starts increasing at 20 ka.

Sinclair et al. constructed probability density functions of deglacial ^{10}Be ages (caption of their Figure 4). They use “ ^{10}Be ages from Greenland to document the spatial and temporal patterns of retreat of the Greenland ice sheet during the last deglaciation” (from their abstract). We interpret this to mean that the ^{10}Be samples they have used were exposed due to glacier retreat. Glacier retreat index is a term we created but have now removed to avoid confusion. We have replaced “glacier retreat index” with “deglacial ^{10}Be ages”.

Page 11, lines 18-20. This statement just isn't true. As a glacier recedes, its surface lowers exposing valley walls at a similar time as its terminus retreats exposing outwash.

We had sea terminating glaciers in mind. This has now been deleted.

Page 11, lines 23-24. What is meant by the "relatively large statistical uncertainty of the large particle concentrations"? This is the first mention of a large uncertainty in the data. If this is an issue it should be discussed clearly in the first reporting of the results and at that point limitations should be set on interpretations.

There is a larger uncertainty on data younger than 7.5 ka BP because it has not been constrained by Coulter Counter data (see Figure 3). Hence the word "relatively". We have added a reference to the Methods section.

Page 11, lines 26-27. What is meant by "a subsequent readvance"? What is this based on?

It was based on our interpretation of the Nielsen et al. modelling paper. In line with our response to a previous comment, it has been deleted.

Page 11, lines 29-32. I think this is the big story here. It's the exposure of the large outwash plains with relative sea level fall. This is interesting and I think would be better to focus on this rather than other less well supported points above.

We thank the reviewer for bringing this to our attention. While we had already discounted the possible influence of exposed continental shelf areas (addressed by reviewer 3), we had not previously considered the possible influence of changes in sea level within the Scoresbysund fjord system.

We have now expanded the relevant sections (abstract, introduction, discussion) to note that the combined product of glacier retreat and advance and relative sea level change likely produce the observed local dust emission changes.

We have also attempted to identify the source of RECAP dust by sampling exposed terrain in low-altitude areas of Scoresby Sound. Unfortunately none of the samples collected is a perfect match to the RECAP data but of the samples, the two that are most compatible were obtained from the foot of Renland peninsula (Fig 4, upward-pointing triangle) and from Jameson Land (Fig 4, square). On this basis we consider the likely source of RECAP dust to be an outwash plain located to the North of Renland (this location is also more compatible with local wind regimes). On the basis of the Sr-Nd data, we cannot discount a contribution from exposed high-altitude regolith within the immediate vicinity of Renland ice cap, but such a source would not be consistent with the timing of activation and deactivation of large particle emissions at the deglaciation and glacial onset, respectively.

Page 12, line 1. What is the isotopic signature of the large particle dust during the last glacial period? This would be interesting to determine as a test of whether the particles originated from Jameson Land.

As explained previously, these measurements have unfortunately not been possible due to the very low concentration of large particles during the glacial.

Page 12, lines 7-10. This is an interesting question about Jameson Land. I think it is reasonable to say that the large particle dust concentration during the last glacial period leaves open the possibility that Jameson Land was ice free, but it certainly doesn't prove it. Again, it would be interesting to do the isotopic signature of this dust to test the source.

If it is from Jameson Land, I'm surprised it doesn't change during stadial-interstadial changes because I would expect that region to be frozen/covered with snow during stadials.

We agree and unfortunately the additional Sr/Nd data collected cannot adequately resolve this question. The reason we consider the possibility of Jameson Land is that Funder et al. 2011 reports that all of East Greenland except for Jameson Land was ice covered. However, our data show only that the dust is local, we do not know the exact source. We have now modified our statement to "The presence of large particles throughout the glacial in the RECAP record is consistent with the existence of local sources throughout the glacial. An ice free Jameson Land cannot be excluded as a possible dust source. "

Also, do the possible source areas change for large particle dust during the last glacial period (as compared to, for example, the Holocene) since winds were likely much stronger during the glacial period?

There is reasonable evidence that winds were stronger on a hemispheric scale during the glacial (eg Schupbach Nature Comms 2018 and references therein) but it is unclear if local winds within Scoresby Sound were significantly greater. Even if we consider a doubling of the atmospheric lifetime of large particles (from 2 to 4 days) it is unlikely that sources from outside of East Greenland could be considered possible.

Page 12, line 11. This is an awkward statement here. It seems like “Here we propose a...” should be in an abstract or at least in the beginning of a manuscript, not in the final paragraph. Also, I don’t necessarily agree that it is a “local paleo glacier extent” proxy. I would suggest it is a relative sea level proxy. As with the previous comments, we agree and the relative sea level has been emphasised now.

Reviewer #3 (Remarks to the Author):

This is an interesting and clearly written manuscript. It claims to demonstrate that large dust particles in the RECAP glacial ice originate from local glacier outwash plains in East Greenland. This is novel and finding the dust source is certainly of broad interest to the scientific community. The source, transport and deposition of fine dust on GrIS is relatively well understood whereas less is known about the source of large particles. Clearly, their size does not allow long transport or residence time in the atmosphere, so a local/regional source is likely. It makes fully sense to look for a source in Greenland. I have some concerns that must be addressed before I could potentially be convinced completely about this new proxy for local glacier extent in Greenland. They are:

P. 2: "During the last glacial period, the ice core dust concentration was 10-100 times greater than in the Holocene due to enhanced continental aridity, increased wind strength and longer atmospheric particle lifetime [7-9]."

Are there any studies or data that show what effect the exposed continental shelves and shallow seas had as dust sources? Including the shelves around Greenland.

The question of dust emissions during the glacial from exposed continental shelves has been discussed in the literature. There are no samples of shelf material available to either support or reject this hypothesis, but the timing of dust variability does not match the timing of shelf exposure and the Sr/Nd/Pb isotopic signature of the dust can be explained by present-day active dust sources in Central Asia. Therefore there is little need to invoke additional dust sources to explain the data. To quote Svensson (JGR, 2007): 'If the shelf areas were a significant source during the last glacial period, the mineralogy, the isotopic composition of Sr and Nd, and the REE composition of these areas must have been very similar to that of the East Asian dust. Considering the variability in tracer characteristics determined for various potential source areas, this possibility seems highly unlikely, although it cannot be ruled out until PSA samples from the shelves have been analysed.'

P. 3: "It furthermore shows that outwash plains became inactive 111.0 ± 0.4 ka b2k during the post Eemian ice sheet advance."

Why did the outwash plains not become larger and more active when the glaciers advanced? At this time sea level was dropping so more fjords and continental shelf was being exposed (and could act as dust sources). The outwash plains can build out in the fjords as the sea level drops.

Glacial isostasy is a non-linear process. We have made a small schematic (Figure C, at the end of this document) to assist in explaining the following description:

At the glacial onset, as glacier volume increased, the absolute sea level dropped. The increased weight of the ice depressed the underlying rock via isostatic adjustment. Relative sea level is the difference between the absolute sea level and the isostatically adjusted crust level. This isostatic adjustment was delayed relative to the sea level decrease, so the relative sea level dropped at the onset of the glacial. However, eventually, the isostatic adjustment was stronger than the absolute sea level decrease, so the relative sea level increased to a higher value (around 70 m according to Lecavalier et al. QSR, 2014) during the glacial. We argue that this is the process leading to the disappearance of dust sources in the Scoresby Sund region. The exact timing and duration of the period of relative sea level decrease at the glacial onset cannot be determined from our data but we can say that it ended before 113 ka BP.

P. 5, Figure 2: Why do the glacial data only cover the period from 12,800-33,900 y b2k? And why do the Holocene data only cover the period 356-4,010 y b2k? Are there data for the whole glacial and the whole Holocene? And why exactly these periods? What if other periods were selected or if the complete glacial and Holocene were studied? The results would be much more convincing if they included the whole periods.

Reviewer #2 also commented on this. We replied: "The samples were originally measured as part of a preliminary study before we started this present work (published in Simonsen et al., Clim. Past, 2018). They were chosen based on sample availability, and no measured samples have been omitted for this manuscript. The samples can all be seen in the Appendix E "Coulter Counter ages", and a reference to this appendix has been added to the caption of Figure 2. We have added two lines on the choice of samples to the Methods." and "We agree with the reviewer that it would be ideal to sample the whole Holocene, but both sample availability and laboratory/instrument time precluded this. We note that previous publications of dust particle size distributions in Greenland ice have found no significant variability in particle size mode between ice core locations (GRIP, NGRIP) and only a minor shift from glacial to interglacial (eg Steffensen et al., JGR 1997; Ruth et al. JGR, 2003). Thus we felt that the measurements presented are reasonably representative of the Holocene dust size distribution at RECAP. We have compared particle size distributions at different time periods of the Holocene and glacial. These are shown in Figure A at the end of this document.. This,

together with the flat large/small particle ratio in the early Holocene (Figure 3), suggests that the size distribution is the same over the Holocene.

”

P. 7: “The relatively constant large particle flux to RECAP over stadial/interstadial cycles suggests that the activity of such local dust sources is not greatly affected from stadials to interstadials.
OK, very good. The data are convincing regarding this.

P.10: “However, exposure dates are typically measured on the valley sides which were exposed before the outwash plain [29].”

No. And this is not generally the case for CN datings such as the wording suggests. The ages from Kelly et al. (2008) are both from moraines on the valley floor and from moraines along the valley sides. The glacier front must be expected to retreat in pace with glacier thinning, so I would not expect the valley sides to be exposed prior to the frontal retreat (and, thus, increase of glacier forefield area).

Reviewer #2 also commented on this. We replied: “We had sea terminating glaciers in mind. This has now been deleted.”

P. 10: “with the warmest period of the Holocene Climatic Optimum.”

Add reference documenting that HCO occurred at 8 ka at this site.

Yes, the reference to Vinther et al. 2009 has been added.

P. 10: “Exposure of outwash plains was further enhanced by decreasing relative sea level from 80 ± 10 to 40 ± 5 m above the present level from 11.0 to 9.0 ka b2k [31]. As most of Schuchert Dal is less than 50 m above sea level, the relative sea level lowering during the deglaciation would have greatly influenced the area of the outwash plain.”

The opposite argument must then also be valid – i.e. when RSL lowers at the onset of the last glacial period (after the Eemian), the area of outwash plains increases.

See above for a discussion of post Eemian relative sea level.

P. 10: “lack of glacial erosion younger than the previous glacial (Saalian) suggests that it was ice free [1,33,34].”

Could as well be explained by a cover of thin, cold-based ice. Why do you think that is not a likely scenario? There is ongoing debate about the status of ice cover over Jameson Land during the glacial and we do not discount either scenario. That is why we write “it is still disputed whether Jameson Land 50 km east of Renland was ice free or not. Young 10Be ages are consistent with an ice covered Jameson Land [32], while lack of glacial erosion younger than the previous glacial (Saalian) suggests that it was ice free [1,33,34].”

P. 10: “points to the existence of exposed land despite glacial coverage of major dust sources.”

What about the continental shelf E of Greenland? In particular the inter-trough areas.

In a previous comment we have discussed our basis for considering continental shelves an unlikely dust source during the glacial.

We have also discussed this with Jan Erik Arndt from the Alfred Wegener Institute, who wrote the recent paper “Marine geomorphological record of Ice Sheet development in East Greenland since the Last Glacial Maximum” in Journal of Quaternary Science, 2018. He says regarding the ice sheet extent during the last glacial maximum: “in my reconstruction I am mainly looking at the marine areas and its geomorphology. These indicate that (at least in the cross-shelf troughs) the Greenland ice sheet extended to the continental shelf break. I guess the shallow banks in between would be covered by ice domes but it is tough getting reliable data for this. But even if not covered by ice domes, these areas would be to a huge part below sea level.”

So it is unlikely that there were any exposed shelves during the last glacial maximum. As the large particle flux is constant over the glacial, there is no reason to believe continental shelves should have been the dominating source during earlier parts of the glacial.

We have amended the text for clarity: “During the glacial, the low but nonzero dust content points to the existence of some exposed land and therefore suggests incomplete deactivation and/or coverage of interglacial dust sources.”

P. 22: “Satellite images give direct evidence of dust storms in the Scoresby Sund region (Figure 10).”

The image does not show a dust storm, it is a view of a sediment-laden glacial meltwater plume flowing into the fjord on a good weather day. Icebergs can be observed in the fjord. Sediment plumes are seen in the water, not in the air! This is commonly observed at such deltas. Does the meteorological data support that there was a storm on August 12, 2012? As far as I can see, the Scoresbysund weather station was out of order that day but Danmarkshavn and Tasiilaq measured high pressure and slow winds. That does not exclude a local storm on this day in Schuchert Dal, but please check up on this.

We have replaced the image by one with a more clearly visible dust storm. Our motivation in showing the image is primarily to indicate that conditions in the local area do support dust deflation during the Holocene. Furthermore we wish to alert the community to a previously unrecognised high-latitude dust source region.

The ice core analyses, data and statistical treatment appear very solid and sound. The reference to ice core literature is good, but the paper is relatively weak when it comes to linking it to surface processes and glacial geomorphology. An example is the use of "outwash plain". Not all land that becomes exposed by glacier retreat is an outwash plain. Much of it would typically be till plains (moraines) consisting of unsorted sediments (diamict) less prone to erosion by the wind than outwash plains. Similarly, the land that becomes exposed by RSL lowering (regression) is a completely different surface than the outwash plains characterized by braided rivers.

We thank you for these constructive comments and for pointing out the weaker aspects of the manuscript. We have clarified that glacial advance and retreat is inextricably connected with RSL change and that both processes contribute to the dust record from Renland. We have shown that outwash plains in the Scoresby Sund region are a source of large dust storms. However, it is beyond the scope of this work to provide a conclusive identification of the exact local source of the large particle dust found in RECAP Holocene ice.

In conclusion, I think the dataset is very interesting and it could potentially be published here if ALL the issues are addressed. This could mean new data collection and analyses. I am also not convinced that the source of the coarse-grained dust is the outwash plains. At least not based on the attempted modern analogue of a "dust storm" from the satellite image (Fig. 10). I would like to see measurements from sediment traps or snow on the ice cap that can be clearly linked to wind erosion during observed storms of outwash plains near the Renland Ice Cap. This should be possible, and the phenomenon is well known from Iceland where the albedo of the ice caps can be dramatically changed during summers where dust storms erode the outwash plains and other loose sediment surfaces. This is most likely documented on satellite imagery (in E Greenland, too). Are there other potential coarse-grained sources in the landscape? Are there any high-altitude plateau in the mountains that might act as a source area?

We appreciate that the reviewer finds the data of interest and that the manuscript has potential for publication. Our goal in this manuscript was not to identify the specific source of Renland dust, but to define the source area within Kong Christian X Land and possibly within Scoresby Sund. Furthermore we were interested to report for the first time that large particles are observed in interglacial ice from a Greenland ice core and that this allows chronological constraints to be applied to the growth and retreat of the ice sheet and local glaciers.

We did not have the opportunity to conduct year-round sampling of dust in Scoresby Sund (this was the topic of a recently rejected grant application) but in October 2018 we did collect surface samples from several locations within Scoresby Sund which have been analysed for Sr/Nd isotopic compositions. These measurements have been included in the revised text (with additional co-authors). Unfortunately, it was not possible to sample dust directly from Schuchert Dal nor from outwash plains located upwind (ie, to the North) of Renland ice cap. From the available data, we have revised our interpretation to put forward the hypothesis that Renland dust likely originates from locations to the North of Renland peninsula as well as possibly from high-altitude exposed regolith in the immediate vicinity of the ice cap. These types of dust emission sources are the most likely based on our recent Sr/Nd isotopic investigations, which have been added to the revised manuscript.

With regard to analogs in Iceland, it is important to note that the mass of large dust particles deposited to Renland is orders of magnitude less than that deposited on Vatnajökull and is certainly too small to be quantified by dust traps or observed by satellite. Again our use of satellite images is to demonstrate that dust deflation processes are active in the Scoresby Sound region, considering that the region has never previously been studied in this context.

Minor issues:

P.1. "past extent of Greenland..." , not paleo. This appears several places in the text.

OK

P. 2. "sea shells on raised beaches". Should be "marine mollusc shells" and note that one would prefer non-marine material for such analyses, e.g. driftwood. See e.g., Funder et al. (2011) in Science.

OK, we have replaced this by "organic material".

P. 4. "stable water isotopes". Should be either stable oxygen or hydrogen isotopes.

Thanks. They actually used atmospheric delta15N, so we have changed the caption to “reconstructed from atmospheric nitrogen isotope ratios”

P. 6. In the caption to Fig. 3, add “(green)” to an explanation of the green curve in sub-figure d.

Yes, done.

P. 8. “Glacial and Holocene dust found in Central Greenland ice cores have been consistently attributed to Central Asian dust sources [12, 23].” This is a repetition – delete.

Yes, deleted.

P. 23. “Figure 9” should be “Figure 10”.

There are 9 figures in the manuscript now.

References:

Alley R.B. and B.R. Koci, Ice-Core Analysis at Site A, Greenland: Preliminary Results, *Annals of Glaciology*, 1988

Arndt, J.E., Marine geomorphological record of Ice Sheet development in East Greenland since the Last Glacial Maximum, *Journal of Quaternary Science*, 2018

Funder, S., The Greenland Ice Sheet During the Past 300,000 Years: A Review, *Developments in Quaternary Sciences*, 2011

Lecavalier, B.S. et al., A model of Greenland ice sheet deglaciation constrained by observations of relative sea level and ice extent, *Quaternary Science Reviews*, 2014

Ruth, U. et al., Continuous record of microparticle concentration and size distribution in the central Greenland NGRIP ice core during the last glacial period, *Journal of Geophysical Research*, 2003

Schupbach, S. et al., Greenland records of aerosol source and atmospheric lifetime changes from the Eemian to the Holocene, *Nature Communications*, 2018

Steffensen, J.P., The size distribution of microparticles from selected segments of the Greenland Ice Core Project ice core representing different climatic periods, *Journal of Geophysical Research*, 1997

Svensson, A., P.E. Biscaye, F.E. Grousset, Characterization of late glacial continental dust in the Greenland Ice Core Project ice core, *Journal of Geophysical Research*, 2000

Vinther, B.M. et al., Holocene thinning of the Greenland ice sheet, *Letters to Nature*, 2009

Figure A

Figure B

Figure C

M = DIFFERENCE BETWEEN
ISOSTATIC CRUST LEVEL AND
ABSOLUTE SEA LEVEL
= RELATIVE SEA LEVEL

Reviewers' comments:

Reviewer #2 (Remarks to the Author):

I reviewed this paper before and it has been improved but I still find some problems. The dataset is valuable and exciting, I and would like to support its publication in a journal like Nature Communications, but I am just not convinced that the data support the interpretations/conclusions. Perhaps the authors could take a bit of a different track for presentation/interpretation of the data. However, I understand this is already a re-review.

My main concern is that the authors make broad and important claims that the dust record resolves glacier extents, which it does not. They go as far as saying that the record is an improvement on what is known (which it isn't, except for maybe/somewhat the Eemian), and that it will be valuable as a "tie point for glaciological models". This really isn't supported by the data presented, and the glaciological models need much more detailed information about the actual locations of ice margins, rather than just presence/absence of ice (or the exposure of outwash and marine sediments) in certain places.

Another concern I have is that the new geochemical data don't seem to support the interpretation that the interglacial dust source is local to Scoresby Sund. Perhaps I'm not understanding this correctly (and the text seems to be contradictory – see detailed comments below), but the Sr/Nd data from RECAP and Renland are ~0.74-0.75, whereas the local Scoresby Sund Sr/ Nd data are mostly ~0.71-0.72, with some ~0.73, but all <0.74. The authors don't seem to explain this, just saying that another source must have contributed as well. From where? What does this mean about the overall interpretations of glacial advances/cover in Scoresby Sund?

Finally, I find the paper still somewhat difficult to read because it is not well organized, there are some random points in different places, and there is quite a bit of repetition of some points. I point some of these out below. It would benefit from better structure overall and within the sections.

More detailed comments:

Page 1

Line 22-24 – It really seems to me that it is a stretch to say that the data presented here are going to provide constraints for ice sheet models. Maybe (but probably not) for the Eemian, and certainly not for the Holocene. The data are simply not providing any detailed information about ice sheet extents. Perhaps take a different angle for the introduction of the paper?

Line 27 – What is meant by "local glacier extent"? This term is never used again (I believe) and it's not clear here what it refers to – do the authors mean small ice caps and glaciers or the ice sheet margin or ice sheet outlet glaciers? Or all of the above?

Page 2

Line 16-17. This just isn't true that there are "few measurements available for this region". There has been a lot of work done in the region. There is certainly less known for the Eemian, but there have been a lot of studies that focused on deglaciation and the early Holocene.

Page 3

Line 4 – The introduction to the RECAP ice core/site comes out of nowhere. Suggest moving the sentence in lines 16-17 up to the beginning of the paragraph (line 4) so that the text first says "here we study..." and then introduces the site.

Lines 20-21 – The Holocene is mentioned here but not the Eemian. Why?

Lines 21-23 – I just don't agree with this statement "... provides an important constraint on the timing of past glacier extent in the Scoresby Sund, thereby providing an important tie point for

paleo ice sheet models”.

Page 4

Figure 1 – Since the focus is the Eemian and the early Holocene, it would great to have these parts of the records shown on an enlarged scale in the paper itself (not in a supplement).

Page 5

Lines 5-8 – I don't understand this sentence. Why does the similarity of concentration and distribution suggest that the dust from all of these sites originated from Central Asia? Hasn't this interpretation already been made in other studies for the Central Greenland sites? Are the authors here trying to say that RECAP looks like the Central Greenland sites, so the RECAP dust can be also interpreted to have originated from Central Asia too? Perhaps rewrite to make this clear.

Lines 23-25 – I don't understand how this sentence/point relates to what is discussed in the paragraph. It may be important information, but perhaps would be placed into context better elsewhere in the paper.

Page 8

Lines 5-7 – I don't understand this statement. Does this mean that the glacial changes in large particle concentration in Renland can be explained solely by changes in snow accumulation? If so, what is the significance of this?

Lines 29-31 - The satellite image of the dust storm is really neat but it looks like the dust is transported eastward. This is away from the Renland site, no? What is the evidence that dust from a source like Schuchertdal which is east of Renland would make it back to Renland?

Page 9

Lines 8-12 – This is a point of major confusion for me. As mentioned above, it looks to me like the local Scoresby Nd/Sr are lower than those of dust in RECAP and Renland. In these two sentences, I think the authors say two contradictory things: First (lines 8-10) that the samples are “consistent with a local origin for RECAP dust”, but then second (lines 10-12) that the RECAP dust has ratios greater than measured in Scoresby Sund samples “indicating that other locations of dust deflation most likely contribute to RECAP dust”. Which is it? And if it is the latter, then what other locations could be sources? And what does this mean for the interpretations of glacier extent/relative sea level in Scoresby?

Lines 30-32 – This is the second or third time that it is mentioned that the stadial-interstadial dust fluctuations in RECAP are similar to those in the Central Greenland ice cores. Suggest limiting the mention/discussion of this result to one place in the text and explain it well there.

Page 11

Lines 15-19 – What about the possibility that the dust source was “extinguished” by snow cover/frozen ground? How is it known that is was due to glacier advance/relative sea level rise?

Reviewer #3 (Remarks to the Author):

I think the authors have responded well to the first review round, and the manuscript is significantly improved. The new data support the conclusions. I still find that the weakest point is the proposed relative sea level change (transgression) at 113.4-111 ka, e.g. on p. 11 line 15-20 and elsewhere. The sketch provided in the rebuttal letter is very simplified. It is not clear to me if the authors mean the eustatic sea level with “absolute sea level” in this sketch. Furthermore, I think the authors need to provide more solid support for a major isostatic depression of the crust just after the Eemian. It must be much larger than the eustatic sea level drop in this period in

order to cause transgression in the study area. I am not qualified to assess if the crust is able to respond so fast to an increased ice load, and just in this short time window. The eustatic sea level change is well known in this time window, see e.g. <https://doi.org/10.5194/cp-12-1079-2016> and references therein so the only unknown parameter is the local response of the crust. There might be literature that has explored this and provided data or GIA modelling results on the response time or rate of crustal depression at the onset of the Weichselian. It appears likelier and a simpler explanation that an early Weichselian glacier advance caused a shut down of the dust source.

This is a minor, but still important, issue. I recommend that the authors explore/discuss it in further detail to support their conclusions.

Other than that, I recommend publication.

Anders Schomacker

Responses to reviewers, manuscript NCOMMS-18-33954A
July 10, 2019

Responses to Reviewer 2

I reviewed this paper before and it has been improved but I still find some problems. The dataset is valuable and exciting, I and would like to support its publication in a journal like Nature Communications, but I am just not convinced that the data support the interpretations/conclusions. Perhaps the authors could take a bit of a different track for presentation/interpretation of the data. However, I understand this is already a re-review.

We thank the reviewer for their recognition that the dataset is valuable and exciting and for taking the time to provide a second review. We have thoroughly revised the manuscript to better present our interpretation of the data and to avoid repetition or presentation of unnecessary information. We also provide new geochemical isotope data supporting our revised interpretation of Renland dust sources.

My main concern is that the authors make broad and important claims that the dust record resolves glacier extents, which it does not. They go as far as saying that the record is an improvement on what is known (which it isn't, except for maybe/somewhat the Eemian), and that it will be valuable as a "tie point for glaciological models". This really isn't supported by the data presented, and the glaciological models need much more detailed information about the actual locations of ice margins, rather than just presence/absence of ice (or the exposure of outwash and marine sediments) in certain places.

We have modified the text regarding the interpretation of the data presented and its application to ice sheet modeling. Firstly we acknowledge the abundance of measurements and interpretation already conducted in the field of Greenland glacial margin reconstruction, particularly during the deglaciation. Our findings *complement* rather than *improve on* existing knowledge and provide novel results for the timing of the glacial onset, as noted by the reviewer. Finally with the addition of new geochemical data, we are able to provide much better constraints on the source region of Renland dust, enabling our interpretation to be more focused and of more benefit to the glaciology and geomorphology communities.

More generally, we would like to acknowledge that the previous versions of the manuscript have not been clear in describing that this research contributes to the improvement of millennial-scale reconstructions of Greenland ice sheet extent. We use the term 'extent' in a semi-quantitative sense, regarding the overall position and extension of the eastern margin of the Greenland ice sheet. Our findings are suitable for models capturing the multi-millennial dynamics of the Greenland ice sheet, but not for detailed basin-scale finite-element models of glacier movement (such as e.g. ELMER/ice). Our data do not substitute the cosmogenic dating techniques required to quantitatively map out the margins of individual glacier basins. Instead they complement these techniques by providing highly accurate time constraints on the response of the ice sheet margin to climate forcings. This is of particular value for

timing glacial advances, such as the onset of the last glacial period, which are not well-suited to the use of cosmogenic dating techniques on glaciofluvial features.

Another concern I have is that the new geochemical data don't seem to support the interpretation that the interglacial dust source is local to Scoresby Sund. Perhaps I'm not understanding this correctly (and the text seems to be contradictory – see detailed comments below), but the Sr/Nd data from RECAP and Renland are ~0.74-0.75, whereas the local Scoresby Sund Sr/ Nd data are mostly ~0.71-0.72, with some ~0.73, but all <0.74. The authors don't seem to explain this, just saying that another source must have contributed as well. From where? What does this mean about the overall interpretations of glacial advances/cover in Scoresby Sund?

The reviewer is correct that the geochemical data presented in the previous manuscript version did not adequately account for the location of Renland dust sources, and that we did not explain this well. After a thorough literature search and consideration of wind patterns in the area, we identified exposed rock samples that were suitable candidates and conducted additional Sr/Nd isotope measurements. These new data are presented in Section 3 (and Supplementary table 1) of the manuscript and satisfactorily account for both the Sr/Nd isotope signatures found in RECAP ice and the prevailing winds in central East Greenland.

Finally, I find the paper still somewhat difficult to read because it is not well organized, there are some random points in different places, and there is quite a bit of repetition of some points. I point some of these out below. It would benefit from better structure overall and within the sections.

The text has been thoroughly revised to provide a clear and structured presentation of the data and our interpretations. We thank the reviewer for the following comments, which helped to identify repetitions, omissions and ambiguities.

More detailed comments:

Page 1

Line 22-24 – It really seems to me that it is a stretch to say that the data presented here are going to provide constraints for ice sheet models. Maybe (but probably not) for the Eemian, and certainly not for the Holocene. The data are simply not providing any detailed information about ice sheet extents. Perhaps take a different angle for the introduction of the paper?

We have been more specific about the Renland dust source regions and therefore the specific relevance of this work to defining changes to the position of the ice sheet margin in central East Greenland. Despite a thorough literature search, we were unable to find any previous estimate for the timing of advance of the eastern margin of the Greenland ice sheet at the glacial onset.

Line 27 – What is meant by “local glacier extent”? This term is never used again (I believe) and it’s not clear here what it refers to – do the authors mean small ice caps and glaciers or the ice sheet margin or ice sheet outlet glaciers? Or all of the above?

We have removed all references to local glaciers and now refer consistently to the ice sheet margin in the revised manuscript, in line with the newly identified Renland local dust sources.

Page 2

Line 16-17. This just isn’t true that there are “few measurements available for this region”. There has been a lot of work done in the region. There is certainly less known for the Eemian, but there have been a lot of studies that focused on deglaciation and the early Holocene.

We agree with the reviewer and have amended the text accordingly (lines 46-51). Again, regarding the end of the Eemian/onset of the glacial, we have not been able to find any comparable data.

Page 3

Line 4 – The introduction to the RECAP ice core/site comes out of nowhere. Suggest moving the sentence in lines 16-17 up to the beginning of the paragraph (line 4) so that the text first says “here we study...” and then introduces the site.

We have changed the text as suggested.

Lines 20-21 – The Holocene is mentioned here but not the Eemian. Why?

Our mistake – the text has been amended. The relevant section is now in lines 99-100.

Lines 21-23 – I just don’t agree with this statement “... provides an important constraint on the timing of past glacier extent in the Scoresby Sund, thereby providing an important tie point for paleo ice sheet models”.

The text has been changed in line with the new focus on changes to the ice sheet margin rather than glacier extent.

Page 4

Figure 1 – Since the focus is the Eemian and the early Holocene, it would great to have these parts of the records shown on an enlarged scale in the paper itself (not in a supplement).

We show the Holocene and Eemian sections of RECAP large and small particle concentrations in Figure 4 (Figure 3 in the previous version). We prefer to keep the current format of Figure 1 so as to provide an overview of the glacial and highlight differences between the glacial and adjacent interglacials.

Page 5

Lines 5-8 – I don't understand this sentence. Why does the similarity of concentration and distribution suggest that the dust from all of these sites originated from Central Asia? Hasn't this interpretation already been made in other studies for the Central Greenland sites? Are the authors here trying to say that RECAP looks like the Central Greenland sites, so the RECAP dust can be also interpreted to have originated from Central Asia too? Perhaps rewrite to make this clear.

Section 2 has been thoroughly revised following the suggestions of the reviewer. The text referring to similarities in the various glacial dust records now appears in lines 118-141.

Lines 23-25 – I don't understand how this sentence/point relates to what is discussed in the paragraph. It may be important information, but perhaps would be placed into context better elsewhere in the paper.

The sentence has been moved to lines 195-197 and located in a more appropriate paragraph

Page 8

Lines 5-7 – I don't understand this statement. Does this mean that the glacial changes in large particle concentration in Renland can be explained solely by changes in snow accumulation? If so, what is the significance of this?

The significance of this finding is that the "inactive" state of dust sources local to Renland ice cap was unchanged through the relatively rapid stadial/interstadial (Dansgaard-Oeschger) cycles. This places limits on the movement of the Greenland ice sheet margin through the glacial. The updated text is now in lines 206-209.

Lines 29-31 - The satellite image of the dust storm is really neat but it looks like the dust is transported eastward. This is away from the Renland site, no? What is the evidence that dust from a source like Schuchertdal which is east of Renland would make it back to Renland?

The reviewer is correct that this image was misleading, especially in the context of our new data regarding Renland dust sources. The text has been moved to lines 272-276 and is now used as an example of the likely mechanism of dust deflation in coastal East Greenland.

Page 9

Lines 8-12 – This is a point of major confusion for me. As mentioned above, it looks to me like the local Scoresby Nd/Sr are lower than those of dust in RECAP and Renland. In these two sentences, I think the authors say two contradictory things:

First (lines 8-10) that the samples are “consistent with a local origin for RECAP dust”, but then second (lines 10-12) that the RECAP dust has ratios greater than measured in Scoresby Sund samples “indicating that other locations of dust deflation most likely contribute to RECAP dust”. Which is it? And if it is the latter, then what other locations could be sources? And what does this mean for the interpretations of glacier extent/relative sea level in Scoresby?

The reviewer is correct that the geochemical data presented in the previous version of the manuscript did not adequately explain Renland dust sources. The new data presented here does explain Renland dust sources and the text has been changed to reflect this in lines 245-264. The implications for reconstructing past ice sheet margins and relative sea level are discussed in section 4.

Lines 30-32 – This is the second or third time that it is mentioned that the stadial-interstadial dust fluctuations in RECAP are similar to those in the Central Greenland ice cores. Suggest limiting the mention/discussion of this result to one place in the text and explain it well there.

The text has been moved to section 2 and the discussion is consolidated into the paragraph on lines 118-128.

Page 11

Lines 15-19 – What about the possibility that the dust source was “extinguished” by snow cover/frozen ground? How is it known that it was due to glacier advance/relative sea level rise?

In line with our new understanding of the Renland dust sources we have limited our comments regarding the possible exposure of Jameson Land during the glacial. This is not the main outcome of this manuscript and we now just note that the glacial large dust fraction supports the possibility of a small but exposed area of land in central East Greenland during the glacial.

Reviewer #3 (Remarks to the Author):

I think the authors have responded well to the first review round, and the manuscript is significantly improved. The new data support the conclusions. I still find that the weakest point is the proposed relative sea level change (transgression) at 113.4-111 ka, e.g. on p. 11 line 15-20 and elsewhere. The sketch provided in the rebuttal letter is very simplified. It is not clear to me if the authors mean the eustatic sea level with "absolute sea level" in this sketch. Furthermore, I think the authors need to provide more solid support for a major isostatic depression of the crust just after the Eemian. It must be much larger than the eustatic sea level drop in this period in order to cause transgression in the study area. I am not qualified to assess if the crust is able to respond so fast to an increased ice load, and just in this short time window. The eustatic sea level change is well known in this time window, see e.g. <https://doi.org/10.5194/cp-12-1079-2016> and references therein so the only unknown parameter is the local response of the crust. There might be literature that has explored this and provided data or GIA modelling results on the response time or rate of crustal depression at the onset of the Weichselian.

It appears likelier and a simpler explanation that an early Weichselian glacier advance caused a shut down of the dust source.

This is a minor, but still important, issue. I recommend that the authors explore/discuss it in further detail to support their conclusions.

Other than that, I recommend publication. Anders Schomacker

We thank the reviewer for the helpful suggestions, and we do mean eustatic sea level when we write absolute sea level.

We agree with the reviewer that the combination of bedrock isostasy, ice sheet advance/retreat and relative sea level change likely contributed differently to changing the ice sheet margin at the onset and termination of the glacial. We were unable to find any suitable information in the literature with regard to the isostatic history of the Scoresby Sund/Kong Christian X region of Greenland, but we include these considerations in the text at lines 99-102, 276-280, 315-320 and 338-347.

Reviewers' comments:

Reviewer #2 (Remarks to the Author):

This is my third review of the paper and I find the current version much improved from past versions. In particular, the Sr/Nd data explanation is improved as is the explanation of the local/regional dust sources based on these data. The interpretations of relative sea level and ice-sheet extent are also more in line with the data presented.

In terms of publication, I think the RECAP dust record is exciting but I'm not sure whether it results in new information about what we know about Greenland ice sheet extents. The two aspects of the paper that I consider novel are the findings of 1) little to no ice margin changes during stadial/interstadial changes, and 2) ice extent advance at the end of the Eemian. However, both of these are relatively minor points in the paper. Otherwise, the results confirm what is already known about the timing of deglaciation in Central East Greenland, and what is known (or unknown) about the coverage of certain regions of Central East Greenland (e.g., Jameson Land) by ice during the last glacial period.

I have a few detailed comments below. Some of these are more significant and some are small editorial changes.

Lines 25-26. I still disagree that the results provide evidence for specifically where the ice-sheet margin was located, so I would suggest rewording this first sentence to remove the term "Mapping", since the data don't really provide the ability to do this.

Figure 1 – I understand that the authors would like to show an overview of the record (from the Eemian to the Holocene), but I still find it problematic that they don't show the dust changes during the end of the Eemian and late glacial/early Holocene in detail/blown up. I would strongly recommend including a figure that shows more detail/blown up images of the end of the Eemian and late glacial/early Holocene, even if the figure has to be included in the supplement.

Lines 196-197 – It is interesting that the Holocene RECAP large particle concentration peaks at 7.8 ka b2k. The authors mention this and that it is coincident with the Holocene Thermal Maximum. I'd like to see more explanation of why they think these coincide. Does this dust peak mark the minimum extent of ice? Maximum relative sea level?

Figure 2 – Why are the Holocene samples only from the period ~356-4010 years b2k? It seems like the authors stress the data about dust source during deglaciation. Shouldn't then the samples be included from the time period of interest (i.e., 9-11.7 ka b2k)?

Line 228 - Suggest changing the section 3 header to "RECAP dust source apportionment" to be consistent with header for section 2 and because the data discussed are from the RECAP cores.

Lines 248-250 – Again here, why are the Sr/Nd ratios measured in samples only from the period 4-7 ka b2k? The authors are interested in and specifically focus the discussion on the source material during deglaciation (i.e., 9-11.7 ka b2k), but they don't have Sr/Nd ratios from this part of the core.

Figure 3 – There is a Renland data point marked by a red circle on the plot. Should it also be plotted on the map, or mentioned that it is in the same location as the blue circle?

Lines 316-320 - Are there existing studies that provide information about the timing of ice sheet growth and relative sea level change? I would expect so. It would be great to dig into these to provide more information about this question/point. It would also be great to cite some of these studies here.

Lines 322-335 - Would it be possible to get a few Sr/Nd ratios of the large particles during the glacial? This might give more insight into the question about the source of this dust. The authors somewhat implicate Jameson Land, but is it possible that the dust is simply from the likely vast nunatak area in the mountainous region of Central East Greenland? Is there any real tie to Jameson Land here beyond speculation? I suggest reducing the discussion about Jameson Land (e.g., the biological refugium seems extraneous) and mentioning the possibility of other local sources (i.e., nunataks).

Lines 339-342 – I don't understand the meaning of this sentence – what do the authors mean by "changes to climatic parameters contributing to the Greenland ice-sheet mass balance? Suggest rewriting this to make clear. For example, what specifically about the "reduced aridity of Asian deserts" influences the ice-sheet mass balance?

References – The authors cite two review papers for much of the data about Central East Greenland deglaciation and relative sea level changes. While I understand that the number of references is limited in a journal such as Nature Communications, it would be great to cite some of

the original work done in this region.

Reviewer #3 (Remarks to the Author):

I am happy with the revised and much improved manuscript and recommend publication. I think the authors have clarified the issues from previous review rounds and also provide convincing explanatins in the rebuttal letter.

I found one typo: Reference "39. Möller, P.E.R." should be changed to "Möller, P."

Best regards,

Anders Schomacker

Responses to Reviewer 2

This is my third review of the paper and I find the current version much improved from past versions. In particular, the Sr/Nd data explanation is improved as is the explanation of the local/regional dust sources based on these data. The interpretations of relative sea level and ice-sheet extent are also more in line with the data presented.

In terms of publication, I think the RECAP dust record is exciting but I'm not sure whether it results in new information about what we know about Greenland ice sheet extents. The two aspects of the paper that I consider novel are the findings of 1) little to no ice margin changes during stadial/interstadial changes, and 2) ice extent advance at the end of the Eemian. However, both of these are relatively minor points in the paper. Otherwise, the results confirm what is already known about the timing of deglaciation in Central East Greenland, and what is known (or unknown) about the coverage of certain regions of Central East Greenland (e.g., Jameson Land) by ice during the last glacial period.

I have a few detailed comments below. Some of these are more significant and some are small editorial changes.

Lines 25-26. I still disagree that the results provide evidence for specifically where the ice-sheet margin was located, so I would suggest rewording this first sentence to remove the term "Mapping", since the data don't really provide the ability to do this.

The structure of the abstract follows the recommendation of the *Nature Communications* submission guide, to 'provide a general introduction to the topic and a brief non-technical summary of your main results and their implication.'. It is for this reason that the first sentence doesn't strictly describe the data presented in the manuscript. Nonetheless, we have replaced the word 'Mapping' with 'Accurate estimates of'.

Figure 1 – I understand that the authors would like to show an overview of the record (from the Eemian to the Holocene), but I still find it problematic that they don't show the dust changes during the end of the Eemian and late glacial/early Holocene in detail/blown up. I would strongly recommend including a figure that shows more detail/blown up images of the end of the Eemian and late glacial/early Holocene, even if the figure has to be include in the supplement.

We have added Supplementary Figure S4 in Supplement B. It is a re-scaled version of Figure 1, showing details of the periods 5-25 and 100-120 ka b2k.

Lines 196-197 – It is interesting that the Holocene RECAP large particle concentration peaks at 7.8 ka b2k. The authors mention this and that it is coincident with the Holocene Thermal Maximum. I'd like to see more explanation of why they think these coincide. Does this dust peak mark the minimum extent of ice? Maximum relative sea level?

Following the recommendations of reviewer 3 (Anders Schomacker), this dataset does not allow us to disentangle the individual effects of ice sheet advance/regression and relative sea level rise, but interpretations can be made regarding the outcome of the two combined. We have added a discussion of factors possibly contributing to the RECAP large particle dust peak.

Figure 2 – Why are the Holocene samples only from the period ~356-4010 years b2k? It seems like the authors stress the data about dust source during deglaciation. Shouldn't then the samples be included from the time period of interest (i.e., 9-11.7 ka b2k)?

The samples were originally measured as part of a preliminary study before we started this present work (published in Simonsen et al., *Clim. Past*, 2018; reference #47 in the manuscript). They were chosen based on sample availability, as there is high demand for ice core samples corresponding to the deglaciation.

Line 228 - Suggest changing the section 3 header to “RECAP dust source apportionment” to be consistent with header for section 2 and because the data discussed are from the RECAP cores.

The change has been made.

Lines 248-250 – Again here, why are the Sr/Nd ratios measured in samples only from the period 4-7 ka b2k? The authors are interested in and specifically focus the discussion on the source material during deglaciation (i.e., 9-11.7 ka b2k), but they don't have Sr/Nd ratios from this part of the core.

As for the previous question comment regarding Figure 2, the RECAP core sections made available for isotopic measurements were selected before we started this present work. They were chosen based on sample availability, as there is high demand for ice core samples corresponding to the deglaciation.

Figure 3 – There is a Renland data point marked by a red circle on the plot. Should it also be plotted on the map, or mentioned that it is in the same location as the blue circle?

We have added an explanation in the figure caption ‘The Renland ice core (red circle) was drilled in 1988 at a location less than 2 km from the RECAP ice core (blue circles).’.

Lines 316-320 - Are there existing studies that provide information about the timing of ice sheet growth and relative sea level change? I would expect so. It would be great to dig into these to provide more information about this question/point. It would also be great to cite some of these studies here.

We have expanded this section and added appropriate references.

Lines 322-335 - Would it be possible to get a few Sr/Nd ratios of the large particles during the glacial? This might give more insight into the question about the source of this dust. The authors somewhat implicate Jameson Land, but is it possible that the dust is simply from the likely vast nunatak area in the mountainous region of Central East Greenland? Is there any real tie to Jameson Land here beyond

speculation? I suggest reducing the discussion about Jameson Land (e.g., the biological refugium seems extraneous) and mentioning the possibility of other local sources (i.e., nunataks).

Regarding the measurement of large particle dust during the glacial, this is unfortunately not possible due to the extremely low dust concentrations present in the ice. There is not enough sample available to produce accurate isotopic data.

As the manuscript has developed, it has become clear that any connection between Renland glacial dust and Jameson Land is tenuous and unlikely. We agree with the reviewer and have reduced this section to a mention of the possibility of exposed land during the glacial.

Lines 339-342 – I don't understand the meaning of this sentence – what do the authors mean by “changes to climatic parameters contributing to the Greenland ice-sheet mass balance? Suggest rewriting this to make clear. For example, what specifically about the “reduced aridity of Asian deserts” influences the ice-sheet mass balance?

We have rewritten this section to make it clear that these parameters were geophysical proxies inferred from the deglacial section of the NorthGRIP ice core.

References – The authors cite two review papers for much of the data about Central East Greenland deglaciation and relative sea level changes. While I understand that the number of references is limited in a journal such as Nature Communications, it would be great to cite some of the original work done in this region.

Within the journal format limitations, relevant references have been added.

Reviewer #3 (Remarks to the Author):

I am happy with the revised and much improved manuscript and recommend publication. I think the authors have clarified the issues from previous review rounds and also provide convincing explanations in the rebuttal letter. I found one typo: Reference "39. Möller, P.E.R." should be changed to "Möller, P."

We are happy that the reviewer is satisfied with our changes to the manuscript and recommends publication. The reference has been corrected.

REVIEWERS' COMMENTS:

Reviewer #2 (Remarks to the Author):

I am satisfied that the authors have made the appropriate changes to the manuscript and/or responded to my questions.

Reviewer #3 (Remarks to the Author):

As in last review round, I think the authors have improved the manuscript and clarified the remaining issues raised in the latest review round. I still recommend publication of this interesting and important manuscript. I look forward to see it in Nature Communications.
Anders Schomacker